# Uncovering hidden enhancers through unbiased in vivo testing

Brandon J. Mannion [1,2], Stella Tran[1], Ingrid Plajzer-Frick[1], Catherine S. Novak[1], Veena Afzal[1], Jennifer A. Akiyama [1], Ismael Sospedra-Arrufat[3], Sarah Barton[1], Erik Beckman[1], Tyler H. Garvin[1], Patrick Godfrey[1], Janeth Godoy[1], Riana D. Hunter[1], Momoe Kato[1], Michael Kosicki [1], Anne N. Kronshage[1], Elizabeth A. Lee[1], Eman M. Meky[1], Quan T. Pham[1], Kianna von Maydell[1], Yiwen Zhu [1], Javier Lopez-Rios [3,4], Diane E. Dickel [1,9], Marco Osterwalder [5,6] ✉, Axel Visel [1,7,8] ✉ & Len A. Pennacchio [1,2,7] ✉

Chromatin signatures are widely used to identify tissue-specific in vivo enhancers, but their sensitivity and specificity remains unclear. Here we show that many developmental enhancers remain undetectable using currently available chromatin data. In an initial comparison of over 1200 developmental enhancers with tissue-matched chromatin data, 14% (*n* = 285) lacked canonical enhancer-associated chromatin signatures. To further assess the prevalence of enhancers missed by chromatin profiling approaches, we used a high-throughput transgenic enhancer assay to screen the regulatory landscapes of two key developmental genes at 5 kb resolution, spanning 1.3 Mb of mouse sequence in total. We observed that 23 of 88 (26%) in vivo enhancers discovered by this approach lacked enhancer-associated chromatin signatures in the respective tissue. Our findings suggest the existence of tens of thousands of enhancers that remain undiscovered by currently available chromatin data, underscoring the continued need for expanding resources for enhancer discovery.

The importance of distant-acting enhancers in the temporal and spatial control of human gene expression is well established[1–4]. Proper transcriptional regulation by enhancers, which are particularly enriched near developmentally important genes, enables normal organismal development and function[5–7]. The initial characterization of enhancers was enabled by pioneering molecular studies of individual loci such as locus control regions at the β-globin locus[8–10], the availability of initial noncoding comparative genomic information from species such as mouse, rat, and pufferfish[11–13], and powerful genomic approaches including ChIP-chip and subsequent next-generation sequencing techniques[14–16].

Dedicated genomic efforts such as ENCODE have sought to systematically identify enhancers via suitable in vitro and in vivo approaches[17]. Remarkably, while the human genome contains only ~ 20,000 protein-coding genes, these studies identified on the order of one million putative enhancers[18]. For example, one ChIP-seq study that examined the enhancer-associated mark H3K27ac on a panel of 12

[1]Environmental Genomics & System Biology Division, Lawrence Berkeley National Laboratory, 1 Cyclotron Road, Berkeley, CA, USA. [2]Comparative Biochemistry Program, University of California, Berkeley, CA, USA. [3]Centro Andaluz de Biología del Desarrollo, Consejo Superior de Investigaciones Científicas, Universidad Pablo de Olavide, and Junta de Andalucía, Seville, Spain. [4]Universidad Loyola Andalucía, School of Health Sciences, Seville Campus, 41704, Dos Hermanas, Seville, Spain. [5]Department for BioMedical Research (DBMR), University of Bern, Bern, Switzerland. [6]Department of Cardiology, Bern University Hospital, Bern, Switzerland. [7]U.S. Department of Energy Joint Genome Institute, 1 Cyclotron Road, Berkeley, CA, USA. [8]School of Natural Sciences, University of California, Merced, Merced, California, USA. [9]Present address: Octant, Inc., Emeryville, CA, USA. ✉e-mail: marco.osterwalder@unibe.ch; AVisel@lbl.gov; LAPennacchio@lbl.gov

tissues isolated from eight mouse developmental stages covering critical phases of mammalian prenatal development (embryonic day [E] 10.5 to postnatal day [P]0) uncovered ~200,000 candidate enhancers[19]. Chromatin accessibility (mapped by ATAC-seq) and the histone modifications H3K4me1 and H3K27ac (mapped by ChIP-seq) are widely utilized as canonical enhancer-associated chromatin marks[16,20–22]. However, the accuracy and practical utility of these data sets critically depend on the correlation of the marks examined with true in vivo activity, which can be assessed in transgenic reporter assays[23]. For instance, enhancer validation efforts using in vivo mouse reporter assays revealed a substantial number of false-positives in these putative H3K27ac-derived putative enhancer datasets[19]. Conversely, it remains unknown if the use of these canonical enhancer-associated chromatin marks comprehensively captures all in vivo enhancers or misses substantial numbers of bona fide in vivo enhancers (*i.e.*, false negatives)[24].

To assess the prevalence and characteristics of enhancers potentially missed in current chromatin-based datasets, we first performed comparisons of pre-existing large functionally validated enhancer sets with comprehensive mouse embryonic tissue chromatin atlases. These retrospective analyses provided initial indications that many in vivo enhancers are missed by chromatin-based discovery strategies that rely on canonical enhancer-associated chromatin marks[19,25]. Next, we conducted a comprehensive prospective analysis in which we performed nearly 300 transgenic enhancer assays[26] for the unbiased tiling of over 1.3 Mb of the mouse genome, which uncovered dozens of hidden enhancers (*i.e.*, without detectable canonical enhancer-associated chromatin marks) in these regions.

## Results

### Many in vivo enhancers show no canonical enhancer marks

As an initial exploration of the comprehensiveness of chromatin-based enhancer mapping strategies, we used the VISTA Enhancer Browser database (https://enhancer.lbl.gov)[25] to retrospectively assess the relationship between enhancer-associated chromatin marks and validated enhancer activity in vivo. To date, this resource includes over 3200 human and mouse elements that have been tested for enhancer-reporter activity, primarily at mouse embryonic day 11.5 (E11.5), a stage when multiple developing tissues (e.g., limb, heart, brain, craniofacial structures) can be assessed through whole-mount imaging in mice and compared with their functional counterparts in humans. We focused on the 1272 validated enhancers that drove reproducible expression in one or more of the following anatomical structures: forebrain ($n = 450$ enhancers); midbrain (398); hindbrain (366); craniofacial region (261); limb (304); and heart (272). We compared these data to chromatin data (H3K27ac ChIP-seq, H3K4me1 ChIP-seq, and ATAC-seq) from these same tissues collected from E11.5 mouse embryos (Supplementary Data 1). For each of the six tissues, we examined the presence of canonical enhancer-associated chromatin signatures at each positive element's endogenous site (Fig. 1a, b and Supplementary Data 2).

For example, for the 304 VISTA limb enhancers, we found that 116 (38%) do not have a limb-specific H3K27ac enhancer-associated mark (Fig. 1c and Supplementary Fig. 1). In addition, of these 116 limb enhancers lacking H3K27ac marks, 60 (20%) also lack an H3K4me1 mark. Finally, 45 of these limb enhancers (15% of VISTA limb-positive elements) are completely lacking any of the three enhancer-associated chromatin marks (H3K27ac ChIP-seq, H3K4me1 ChIP-seq, or ATAC-seq) in limb tissue. Across all six tissues examined, these hidden enhancers represent 9% to 25% of VISTA enhancers (Supplementary Figs. 1–3). Overall, we found that 50% (1028) of tissue-specific VISTA enhancers have all three marks, 22% (461) have at least two marks, 13% (277) have only one of the three marks, and 14% (285) are hidden enhancers without any of the three marks in the corresponding tissue (Fig. 1d and Supplementary Table 1). The relative proportions of these chromatin mark categories are similar across the six considered tissues

(Supplementary Fig. 3). We observed hidden enhancers in all six developing tissues that we assessed at E11.5, both for their enhancer-associated marks and transgenic enhancer-reporter activity, which suggests their existence is a general phenomenon across other cell and tissue types.

### Mouse in vivo tiling assay uncovers additional hidden enhancers

Since many of the enhancers reported in the literature and VISTA database were found through chromatin signature-guided enhancer discovery screens, retrospective intersections are likely to underestimate the proportion of enhancers lacking canonical chromatin signatures. To assess this phenomenon in a more unbiased manner, we selected two separate loci (*Gli3* and *Smad3/Smad6*) to test the enhancer activity of 281 overlapping elements regardless of their chromatin state. The *Gli3* gene encodes a transcription factor that is involved in pathways for the development of the limb, face, and nervous system[27–29]. Apart from *Gli3* itself, the flanking region included in the tiling is generally depleted of other genes and includes a gene desert that spans over 800 kb[30]. Dozens of regions ($n = 38$) across the locus are predicted to be enhancers based on tissue-specific H3K27ac (Fig. 2a and Supplementary Data 3), and prior limited candidate enhancer studies within this locus identified enhancers active in the limb and brain in E11.5 mouse embryos[31–33]. In addition, we performed unbiased tiling across a second locus that encompasses the *Smad3* and *Smad6* genes (Supplementary Fig. 4). As with the *Gli3* locus, the *Smad3/Smad6* locus considered for tiling also includes several ($n = 86$) H3K27ac-marked regions (Supplementary Data 3). While *Smad3* is broadly expressed in all six of the tissues examined in this study at E11.5, expression of *Smad6* is highest in the heart of mouse embryos at E11.5 (Supplementary Fig. 5), consistent with its importance in cardiovascular development[34]. We designed elements spanning ~5 kb in size with boundaries chosen to fully capture complete H3K27ac-enriched regions where possible and with overlaps to adjacent elements in order to tile across both loci. Altogether, we tested 281 of the sequences in a site-directed mouse in vivo transgenic assay[26,35] and assessed enhancer activity in six tissues, for a total number of 1,686 enhancer-tissue observations (Fig. 2b–d). Collectively, the tested elements span over 1.3 Mb (approximately 1 and 0.3 Mb of the *Gli3* and *Smad3/Smad6* loci, respectively) of the mouse genome.

We observed that 63 of 281 tested elements showed reproducible enhancer-reporter activity at mouse embryonic day 11.5 (E11.5) in at least one tissue (Fig. 2b and Supplementary Fig. 4). A majority of elements tested in the *Gli3* locus showed reproducible LacZ activity in the developing brain, limb, and craniofacial regions (Supplementary Fig. 6), tissues in which *Gli3* is expressed[27–29]. Similarly, elements tested around the *Smad3/Smad6* loci show activity in a variety of developing tissues (Supplementary Data 2), which likely reflects both the observed broad expression patterns of *Smad3* and the known tissue-specific roles of *Smad6* in cardiovascular development[34].

Similar to the retrospective VISTA study, we focused on six tissues (forebrain, midbrain, hindbrain, craniofacial structures, limb, heart) to assess the relationship between experimental enhancer data in transgenic reporter assays and chromatin data (H3K27ac, H3K4me1, ATAC-seq, Supplementary Data 2). The 63 elements that showed reproducible in vivo enhancer-reporter activity in one or more developing tissues altogether represented 88 tissue-enhancer activities. We used these 88 tissue-enhancer activities to compare with stage- and tissue-matched chromatin data and observed that 23 (26%) were hidden, *i.e.*, they lack enhancer-associated chromatin marks in their active tissue. We observed that hidden enhancers from the unbiased tiling represent a larger proportion of tissue-specific enhancers (26%) relative to the retrospective VISTA enhancer comparison (14%) described above (Fig. 3a). We identified hidden enhancers in all 6 tissues under investigation, including forebrain (of 15 forebrain-enhancers, 2 were hidden) and hindbrain (of 17 hindbrain-enhancers, 7 were hidden) (Fig. 3b

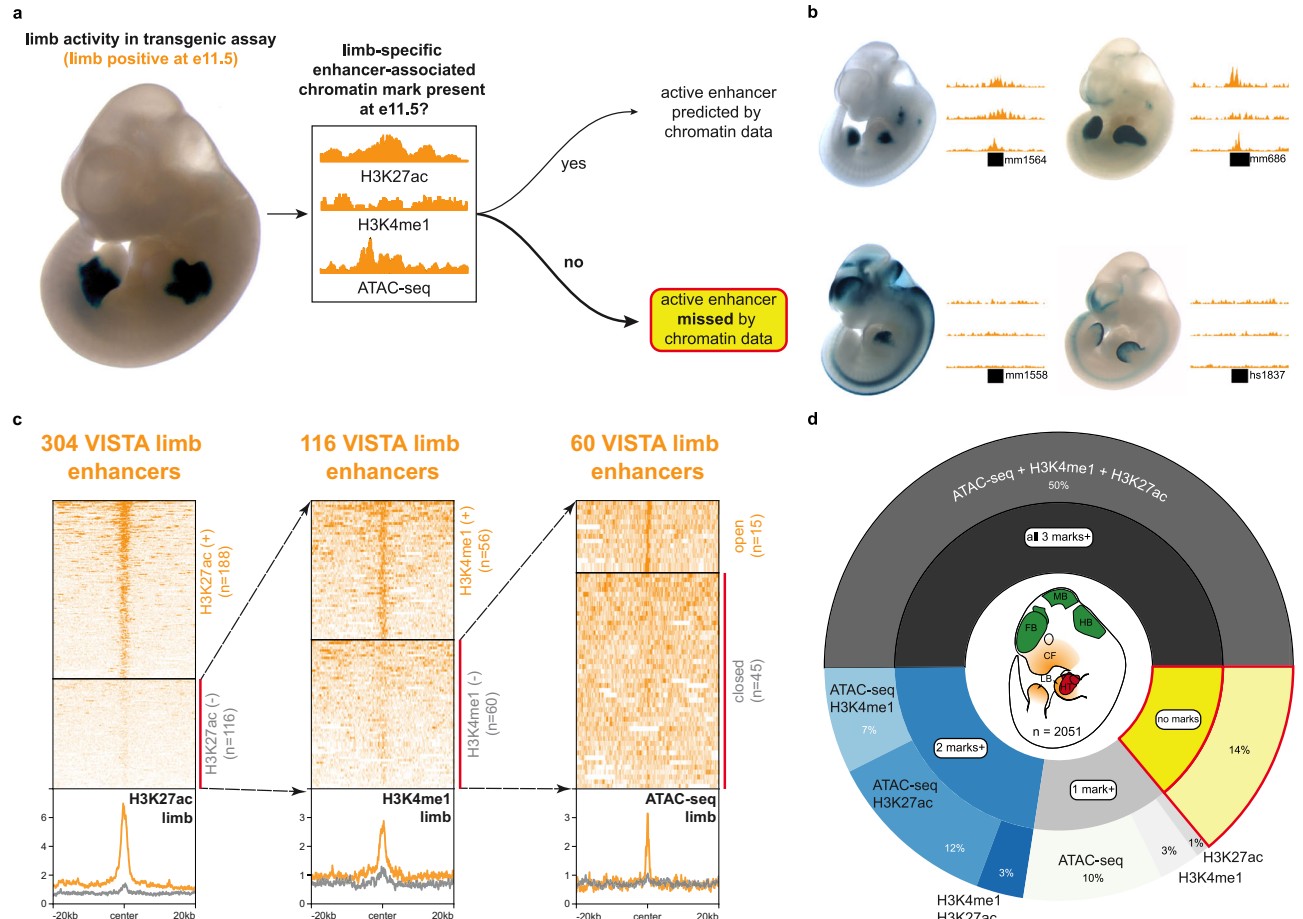

**Fig. 1 | Mouse in vivo enhancers without canonical enhancer-associated chromatin marks. a** Approach to retrospectively identify active enhancers without tissue-specific enhancer-associated chromatin marks. **b** Examples of active limb enhancers with (top row) and without (bottom row) enhancer-associated chromatin marks in stage-matched embryonic limb tissue. See Supplementary Fig. 1 for examples of active enhancers without these marks in other tissues. **c** Chromatin profiles of active limb enhancers with and without H3K27ac (ChIP-seq), H3K4me1 (ChIP-seq), or open chromatin (ATAC-seq). See Supplementary Fig. 2 for another example of chromatin mark filtering for forebrain enhancers. **d** Proportion of VISTA enhancers across six tissues (forebrain, midbrain, hindbrain, craniofacial structure, limb, heart) with and without enhancer-associated chromatin marks. For this study, we focused on the VISTA enhancers with activity ("positive" elements) in the above six tissues (Supplementary Table 1). Active enhancers without any of these chromatin marks are in yellow.

and Supplementary Fig. 6). Aside from a lack of the three enhancer-associated chromatin marks, we found a majority of these hidden enhancers also were without alternative histone marks examined by ENCODE (H3K27me3, H3K36me3, H3K4me2, H3K4me3, H3K9ac, H3K9me3) (Fig. 3b and Supplementary Fig. 6). Further, hidden enhancers across the *Gli3* locus were active in tissues that show *Gli3* mRNA expression in situ and marked enhancers at the same developmental stage, which supports their role in regulating neighboring gene expression (Supplementary Fig. 7). In addition, using CRISPR/Cas9 genome editing, we generated a mouse line encoding a deletion of a 303 kb intra-TAD (iTAD) region upstream of the *Gli3* TSS (Supplementary Fig. 8A and Supplementary Tables 2, 3). The deleted region (iTAD$^d$) encompasses seven canonical and four hidden enhancers, each with distinct tissue-specific activities. Remarkably, homozygous deletion of this interval did not reveal any gross phenotypic changes, and limb morphology remained normal despite the absence of the mm1179 (canonical)[32] and mm2164 (hidden) limb enhancers (Supplementary Fig. 8B, C). While this result was in line with our prior study revealing compensation of individual (mm1179) enhancer function through redundant interactions[33], we leveraged our *Gli3* deletion allele[33] to investigate the functional contribution of the mm2164 hidden enhancer in direct comparison with mm1179 at reduced *Gli3* dosage (Supplementary Fig. 8C, D). For this purpose, we compared limb skeletons of *Gli3*$^{d/+}$, mm1179$^d$/*Gli3*$^d$ and iTAD$^d$/*Gli3*$^d$ embryos. A

split digit 1 phenotype observed in *Gli3*$^{d/+}$ embryos is exacerbated in mm1179$^d$/*Gli3*$^d$ embryos, which show digit 1 duplication[33]. In addition, a subset of mm1179$^d$/*Gli3*$^d$ embryos (n = 4/7) also exhibited bifurcation of digit 2, but only in the right forelimb (RFL)[33]. Upon iTAD deletion (iTAD$^d$/*Gli3*$^d$), which removes both the canonical mm1179 and the hidden mm2164 limb enhancers, we observed a slightly more compromised limb phenotype than in mm1179$^d$/*Gli3*$^d$ embryos (Supplementary Fig. 8D). Specifically, in addition to duplication of digit 1, we observed a substantial increase in the frequency of additional bifurcation of digit 2 in iTAD$^d$/*Gli3*$^d$ embryos that affected predominately the left forelimb (LFL, Supplementary Fig. 8D). In iTAD$^d$/*Gli3*$^d$ embryos, we observed LFL digit 2 bifurcation in 2 of 3 embryos (67%), compared to 0 out of 7 in mm1179$^d$/*Gli3*$^d$ embryos (0%). The observed left-right asymmetry of the phenotype is consistent with known asymmetries in limb development and genetic syndromes affecting predominantly the left or right limb[36]. This result supports that the hidden enhancer mm2164 is functional and likely contributes to the robustness of early anterior *Gli3* gene dosage required for anterior digit control[29,33]. Of those regions across the two loci that were predicted to be enhancers based on tissue H3K27ac data and that were subsequently tested for enhancer reporter activity, we found the validation rate of H3K27ac across the six tissues considered to be 28% (Supplementary Data 3). We found that regions with high-ranked H3K27ac signal (i.e., based on H3K27ac signal enrichment) corresponded with higher validation rates

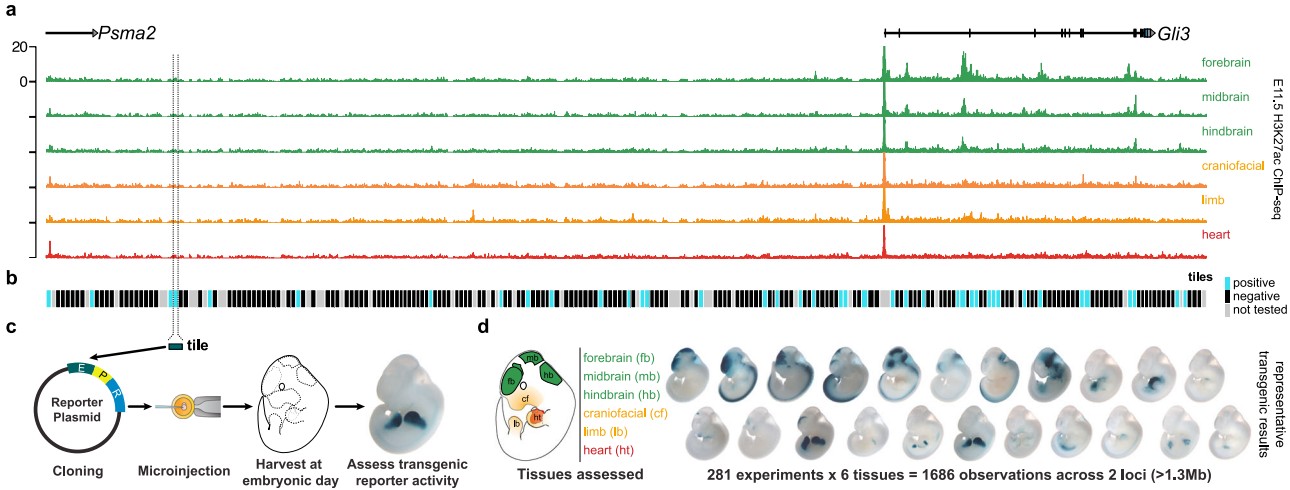

**Fig. 2 | Systematic tiling for the unbiased identification of mouse in vivo enhancers. a** *Gli3* locus with H3K27ac ChIP-seq data (ENCODE) for six tissues. **b** Elements for the unbiased tiling assay were ~ 5 kb in size and designed to overlap with adjacent elements. Elements that were tested and that had reproducible enhancer-reporter activity in the mouse in vivo transgenic assay are shaded blue. We observed 63 tested elements with tissue-specific enhancer-reporter activity at mouse embryonic day 11.5 (E11.5) with tissue-specific activity. Of these 63 enhancers, 36 show reproducible enhancer-reporter activity in multiple tissues at E11.5. Elements without reproducible activity are shaded black. Elements not successfully tested are shaded gray. **c** Approach for testing each tile in the mouse in vivo transgenic assay. E, enhancer; P, promoter; R, reporter. **d** Left: Depiction of tissues that were checked for reproducible enhancer-reporter activity. Right: Example transgenic results from tiling across the *Gli3* and *Smad3/Smad6* loci (see Supplementary Fig. 4 for the *Smad3/Smad6* locus).

than those tested regions with low-ranked H3K27ac signal, which is consistent with previous studies[18,19]. Altogether, our retrospective VISTA study and unbiased systematic experimental testing uncovered 308 tissue-specific hidden enhancers, supporting the existence of substantial numbers of missing enhancers genome-wide.

## Hidden enhancers are indistinguishable from their marked counterparts

Next, we assessed the properties of hidden versus marked enhancers in an attempt to explain their functional differences. From the VISTA retrospective study and the unbiased tiling, both hidden and marked enhancers have similar levels of evolutionary conservation, i.e., both categories have elevated conservation scores (phastCons) relative to genomic background and show no significant difference in the level of conservation (Supplementary Fig. 9). Within each tiling locus we did not find specific transcription factor binding sites that were enriched in hidden enhancers relative to marked enhancers (Supplementary Table 4). Similarly, by functional enrichment analysis, there are no significant biological processes or phenotypes that distinguish hidden enhancers from their marked counterparts (Supplementary Table 5). To explore the potential contributions of transposable elements (TEs) within these enhancer regions[37], we enumerated the TEs within hidden and marked enhancers to assess whether particular TE families were enriched or depleted in either group. We found similar proportions of LINE, SINE, DNA transposon, and other repeat element families between hidden and marked enhancers (Supplementary Fig. 10). These comparisons further confirm that hidden enhancers show all hallmarks of bona fide in vivo enhancers with canonical marks.

## Some hidden enhancers can be identified from additional chromatin data

Given the absence of canonical enhancer-associated chromatin marks in embryonic mouse tissue-derived data, we examined if complementary chromatin data types offer potential avenues for the discovery of these hidden enhancers. We first evaluated if hidden enhancer activity at E11.5 could be the outcome of residual LacZ reporter activity from enhancer activity that occurred at an earlier developmental stage. Of the 308 tissue-specific hidden enhancers assayed at E11.5, 172 (56%) have enhancer-associated chromatin marks at an earlier stage, i.e., H3K27ac and/or H3K4me1 at embryonic day 10.5 (E10.5) (Fig. 4a).

Next, we examined if available single-cell chromatin data could resolve enhancer-associated chromatin marks around hidden enhancers that might have been missed from standard chromatin data derived from bulk tissue preparations. Of the six tissues for which we compared transgenic enhancer-reporter activity with the corresponding mouse tissue chromatin data from ENCODE, for two tissues (forebrain, hindbrain) single-nucleus ATAC-seq (snATAC-seq) data across early mouse development was available[38,39]. Of the 44 hidden enhancers that are active in either the forebrain or the hindbrain, only 8 (18%) could be identified via corresponding single-cell data (Fig. 4b).

Since 235 (82%) of the 285 hidden enhancers identified in the VISTA retrospective study are the human orthologues of human-mouse conserved sequences tested in mouse transgenic enhancer assays, we also examined if available human tissue-matched epigenomic data would have predicted any of these hidden enhancers. For 112 human-derived hidden enhancers that did not have enhancer-associated chromatin marks either in earlier (E10.5) or in single-cell chromatin data, 49 were assessable with available tissue-matched, similar-staged human chromatin data from craniofacial, heart, and limb bud tissues. Only 10 (20%) showed enhancer marks in available human chromatin data. Of the 308 hidden enhancers from either mouse or human sequences, 50 (16%) did not have available single-cell data for their prediction and were not identified either via earlier chromatin data or similar-staged human chromatin data. Altogether through these stage- and tissue-matched analyses, 118 (38%) of the originally identified hidden enhancers could not be identified despite at least two of the three complementary data types being available (Fig. 4c).

Finally, while our major focus has been on comparing experimentally validated enhancers to their chromatin profiles in their exact tissue of activity, we sought to explore if chromatin data from disparate sources (*i.e.*, without the previous constraints to precisely match developmental stage and tissue type to the tissue-specific enhancers of this study) could nonetheless be informative for the identification of hidden enhancers. We used a broad catalog of

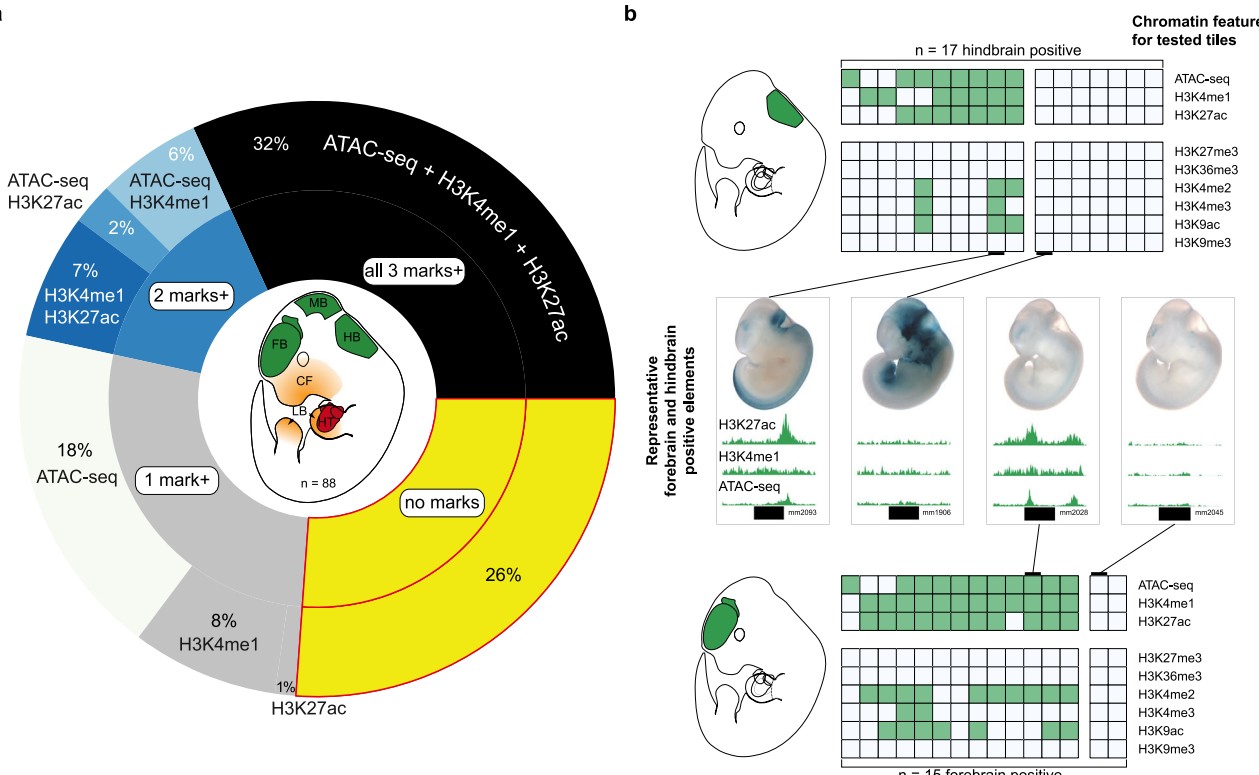

**Fig. 3 | Active enhancers from unbiased tiling with and without enhancer-associated chromatin marks. a** Proportion of active enhancers from unbiased tiling with and without enhancer-associated chromatin marks. Active enhancers without any of these chromatin marks are in yellow. **b** Example of active hindbrain (*n* = 17) and active forebrain enhancers (*n* = 15) from the tiling assay. Columns within the square table represent a tested element (a region with genomic coordinates), whereas rows within a single column represent the chromatin feature (shaded green if a peak in the given chromatin feature is present) for that particular element. The square tables are split into two main categories, those elements with at least one of the considered enhancer-associated chromatin marks present (left;

hindbrain: mm1893, mm2140, mm2124, mm1947, mm1940, mm2019, mm2022, mm1935, mm2093, mm1902; forebrain: mm1879, mm2122, mm2140, mm2019, mm2093, mm2022, mm1935, mm1893, mm1862, mm1912, mm2028, mm1902, mm1977) and those without any of the three considered enhancer-associated chromatin marks (right; hindbrain: mm1906, mm2213, mm2153, mm2134, mm1879, mm1862, mm2152; forebrain: mm2045, mm2134). Additional chromatin data depicted show that a portion of hidden enhancers are not marked by any of the chromatin marks assayed by ENCODE (Supplementary Fig. 6). Representative transgenic results (two enhancers for hindbrain; two for forebrain) are depicted, as well as the chromatin profile for the relevant element.

candidate *cis*-regulatory elements (cCREs) derived from chromatin-based profiling of various human and mouse cell lines and tissues[18]. Collectively, the human- and mouse-derived cCREs annotated for enhancer-like signatures cover over 14% of the mouse genome. Across both the pre-existing VISTA and tiling studies, we found that 243 of 270 (90%) of the hidden enhancers showed enhancer-like signatures (ELS) in at least one data set from this comprehensive cCRE catalog (Supplementary Fig. 11). However, a majority of elements (1233 of 1505; 82%) that were negative in our transgenic mouse assay also overlapped with enhancer-like cCREs. In addition, hidden enhancers that are not marked by cCREs from this expanded search have similar levels of elevated evolutionary conservation as those with cCREs, which further supports their functional constraint (Supplementary Fig. 12). Altogether, intersection with this generalized collection of cCREs from differing cell types and developmental stages has no substantial predictive power beyond the use of tissue-specific chromatin data sets.

## Discussion

In this study, we report the existence of hundreds of hidden enhancers in the human/mouse genome that lack canonical enhancer-associated marks in chromatin profiling data from the tissue in which they are active. This includes a retrospective analysis of over 1200 in vivo validated tissue-specific enhancers in VISTA and a prospective tiling study of 281 candidate sequences, which implemented a recently scaled transgenic assay[26] to systematically test elements for mouse in vivo enhancer activity across over 1.3 Mb of a mammalian genome.

In contrast to previous in vitro approaches or studies in humans, mice, and *Drosophila*[40–45], the present screen represents a comprehensive and systematic assessment of sizable genomic intervals across two mammalian loci for bona fide in vivo enhancer activity. We show that a majority of tissue-specific enhancers have corresponding enhancer-associated chromatin marks in respective tissue(s), which supports the continued use of these datasets for candidate enhancer identification. However, we also show that reliance on currently available chromatin-based datasets to identify candidate enhancers misses a notable portion of seemingly hidden enhancers that are active in the transgenic in vivo reporter assay but do not show any of the noted marks. Within the tiling study across two separate loci, we show that the tissue-specific enhancer-reporter activities of hidden enhancers are similar to those of their marked counterparts, which suggests these sequences contribute tissue-specific enhancer activity at their endogenous sites in ways similar to canonical enhancers. Remarkably, the deletion of a 303 kb intra-TAD (iTAD) *Gli3*-upstream region containing multiple canonical and designated hidden enhancer elements with tissue-specific activities did not result in overt phenotypic alterations, indicating the presence of multiple tissue-specific redundant enhancers outside of the deleted region. This is similar to prior observations at the *Gli3* locus and highlights the challenges in functional enhancer dissection when multiple enhancers can provide redundancy or a buffering effect to maintain gene expression[33]. However, using our previously applied genetic framework for evaluation of *Gli3* enhancer function in a sensitized background[33], we find evidence for the

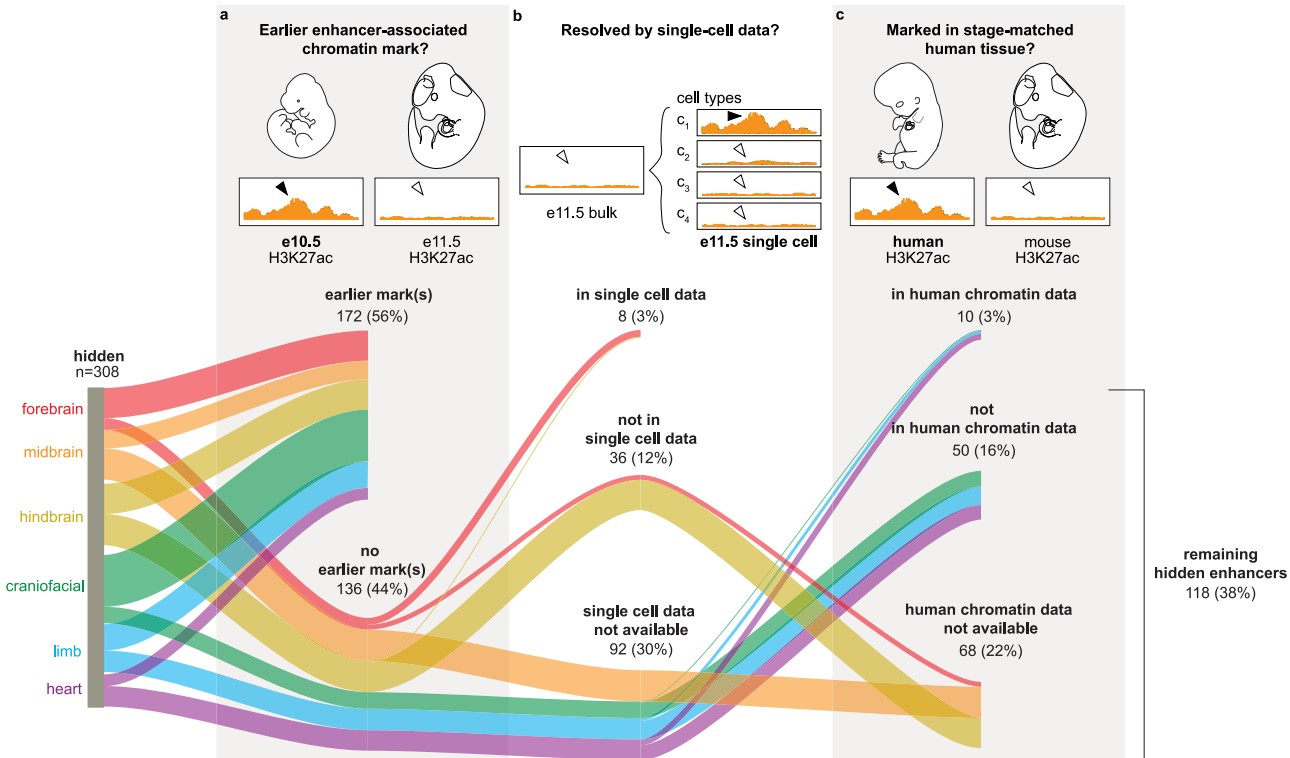

**Fig. 4 | Hidden enhancers cannot be fully recovered from additional chromatin data. a** Hidden enhancers that are active at E11.5 are assessed for earlier enhancer-associated chromatin marks at E10.5. **b** Hidden enhancers based on bulk tissue chromatin data and that do not have earlier enhancer-associated chromatin marks are assessed for corresponding enhancer-associated chromatin marks from available single-cell chromatin accessibility data. **c** Hidden enhancers, neither recoverable from earlier enhancer-associated chromatin marks nor single-cell chromatin accessibility data, are assessed for enhancer-associated chromatin marks in available human chromatin data. Percentages represent the proportion of the filtered elements relative to the starting set of 308 hidden enhancers.

functionality of hidden enhancers. As observed for developmental enhancers of other TF genes, the function of a hidden limb enhancer upstream of *Gli3* (mm2164) appears to contribute to phenotypic robustness[4,33], yet future work to demonstrate roles and potential differences of both marked and designated hidden enhancers in other loci will be required. We also find that the levels of evolutionary conservation between both marked and hidden enhancers are similarly elevated relative to random genomic background, which is also supportive of their functional relevance or utility both within and outside their endogenous contexts. Similarly, forebrain, midbrain, and hindbrain activities were detected most frequently among both hidden and marked enhancer categories, likely related to the complexity of the regulation of brain development[46]. However, apart from the absence of enhancer-associated chromatin marks, we could not identify sequence, genomic, or other epigenomic properties that could distinguish hidden enhancers from their marked counterparts.

We found many of these hidden enhancers can be identified by considering complementary data either from other time points, single-cell chromatin measurements, or other species. Enhancer-associated chromatin marks at an earlier stage (E10.5) identified the largest proportion (56%) of E11.5 hidden enhancers. Reporter systems that provide a higher resolution of in vivo temporal readout of enhancer-reporter activity than the utilized LacZ-based transgenic assay, or functional deletion/inhibition experiments, will determine whether this portion of hidden enhancers exhibit largely stage-restricted functions. Lack of endogenous chromatin context represents another limitation of using site-directed transgenic reporter assays for evaluation of enhancer activity and function. While public single cell chromatin accessibility data are currently limited to a few tissues in mice[38,39], it is likely that additional developing tissues (e.g., face, limb, heart) will soon be surveyed at single-cell resolution. As supported by our comparisons of hidden forebrain and hindbrain enhancers, these single-cell approaches should enable the resolution of both common and rare cell types in tissues and, subsequently, the identification of enhancers missed by bulk tissue-derived data. In addition, human chromatin data within a similar developmental window are currently available only from a few tissues (i.e., heart[47], face[48], limb bud[49]), and future characterization of other similarly staged human tissues should facilitate cross-species comparisons of enhancer-associated chromatin marks and in vivo enhancer activity. Further, we utilized cCREs derived from various human and mouse cell types to recover a majority of the hidden enhancers that were previously missed by the available stage- and tissue-matched chromatin data. Although this raises the possibility that additional chromatin data from other sources can assist with the identification of these hidden enhancers, this is at the expense of poor specificity or selectivity for elements that do validate as active enhancers in the mouse in vivo system. Moreover, these pooled data cannot provide insights into the specific cell type and/or developmental stage of candidate enhancers.

Across the two tiling loci, we found over 80 instances of reproducible tissue-specific enhancer activity, representing ~ 26% of which are hidden enhancers in their corresponding tissue. Although we considered additional chromatin data to identify hidden enhancers, we cannot exclude the possibility that a portion of these active enhancers could instead represent false positives in the transgenic reporter assay rather than false negatives originating from chromatin-based enhancer-prediction datasets. Nonetheless, we cannot yet resolve whether the observation of these hidden enhancers is attributable to a novel biological mechanism or instead to technical limitations of the current data. Of the hidden enhancers identified in this study, it is likely that a majority do have enhancer-associated chromatin marks which were missed due to the presence

of these marks in a limited number of cells or during a time point not captured by these data. We were unable to find any distinguishing properties of hidden enhancers using additional histone modifications profiled in the ENCODE developmental chromatin catalog. In this study, we could not systematically exclude in a tissue- and developmental stage-matched manner the possibility that other histone modifications[50,51] (e.g., H2BNTac) or proteins[22,52,53] (e.g., p300/CBP, Pol II) are present at these enhancers. In consideration of these technical limitations and also their indistinguishability from marked enhancers based on sequence conservation and other genomic properties, we suggest hidden enhancers to be functionally relevant contributors to the regulatory landscapes in which they are present. We focused on only six tissues from which we could compare between tissue-matched chromatin properties and mouse in vivo data at E11.5, yet there are vast numbers of other tissues and developmental time points relevant for enhancer identification[54]. With some estimates of hundreds of thousands to on the order of one million candidate enhancers in mammalian genomes, one might speculate from our tiling study that there are tens of thousands of additional enhancers unaccounted for by current genome-wide chromatin catalogs[18]. As sequencing expands to cover the full range of human tissues, diversity, environmental perturbations, and as related technologies provide even higher resolution approaches to probe gene regulatory activity, we can expect to better understand and annotate the characteristics of enhancers and their functional significance in transcriptional regulation.

## Methods

### Animal studies and experimental design
All animal work was reviewed and approved by the Lawrence Berkeley National Laboratory Animal Welfare and Research Committee under protocol numbers 290003 and 290008. All mice used in this study were housed at the Animal Care Facility (ACF) of LBNL. Mice were monitored daily for food and water intake, and animals were inspected weekly by the Chair of the Animal Welfare and Research Committee and the head of the animal facility in consultation with the veterinary staff. The LBNL ACF is accredited by the American Association for the Accreditation of Laboratory Animal Care International (AAALAC). Transgenic mouse assays and deletion mouse models were performed in the *Mus musculus* FVB/NJ strain. Mouse embryos of both sexes were used in these analyses. Sex was not considered as a variable since limb development is expected to show minimal differences at the respective stages of development.

Mice used for transient transgenic reporter analysis and mice of the *Gli3*[d], mm1179[d], and iTAD[d] lines (strain: FVB/NJ) were housed at the LBNL Animal Care Facility. Mice were maintained with water supply on a 12:12 light-dark cycle, with relative humidity set at 30–70% and a temperature of 20–26.2 °C. Mice were housed in standard microisolator cages on hardwood bedding with enrichment consisting of crinkle-cut naturalistic paper strands and fed on ad libitum PicoLab Rodent Diet 20 (5053). Euthanasia at LBNL was performed in the home cage using $CO_2$ asphyxiation while ensuring gradual fill and displacement rate. Generally, mice between 6 to 30 weeks of age were used for breeding to generate embryos, newborns, or adults analyzed in this study. Sample size selection strategies were conducted as follows:

### Transgenic mouse assays
Sample size selection and sorting criteria were based on our experience of performing transgenic reporter assays in mouse embryos for > 4000 putative enhancer elements (VISTA Enhancer Browser: https://enhancer. lbl.gov). Mouse embryos were excluded from further analysis if they did not encode the reporter transgene or if the developmental stage was not appropriate. For site-directed transgenic reporter assays (enSERT), results were confirmed in at least two independent biological replicates, based on established criteria (see https://enhancer.lbl.gov).

### Knockout mice
Sample sizes were selected empirically based on our previous studies[33,55]. Using a matched littermate selection strategy, embryos described in this study resulted from crossing mice heterozygous for the respective deletion (iTAD[d], Gli3[d] or mm1179[d]) to allow for the comparison of matched littermates of different genotypes. Embryonic littermates and samples from genetically modified animals were dissected and processed blind to genotype.

### CRISPR/Cas9 deletion mouse lines and genotyping
SgRNAs located 5' and 3' of the designated 303 kb *Gli3* upstream deletion (iTAD[d]) were designed using CHOPCHOP[56] (Supplementary Table 2). The iTAD[d] allele was engineered using CRISPR/Cas9 genome editing in fertilized mouse oocytes as previously described[33,35]. A mix of Cas9 mRNA (final concentration of 100 ng/ul) and the two single guide RNAs (sgRNAs) (25 ng/ul each) was microinjected into the cytoplasm of fertilized FVB strain oocytes. High Fidelity Platinum Taq Polymerase (Fisher Scientific) with primers spanning the deletion breakpoint to detect the iTAD[d] allele and flanking the 5' sgRNA region to detect the WT allele were used for identification of F0 and F1 mice (Supplementary Table 3 and Supplementary Fig. 8B). Clean deletion breakpoints in F0 and F1 mice were confirmed using Sanger Sequencing from PCR iTAD[d] amplicons (Supplementary Fig. 8B). *Gli3*[d] and mm1179[d] alleles were genotyped following the same PCR protocol[33].

### ENCODE mouse chromatin and RNA-seq data
Processed mouse chromatin data[19] (ATAC-seq; ChIP-seq for H3K27ac, H3K4me1, H3K4me2, H3K4me3, H3K27me3, H3K9ac, H3K36me3, H3K9me3) and RNA-seq data[57] were downloaded from the ENCODE resource portal (https://www.encodeproject.org/). Details on the generation and processing of these data are available here: https://www.encodeproject.org/pipelines/. See Supplementary Data 1 for a listing of all the bulk tissue mouse data used for chromatin intersections or tissue expression analyses.

### VISTA enhancers
Human and mouse candidate enhancers were tested in a mouse in vivo transgenic reporter assay, as previously described[26] (see also "Locus selection for tiling and mouse in vivo enhancer validation"). Candidate enhancers were assessed for reproducible enhancer-reporter activity in forebrain, midbrain, hindbrain, craniofacial structures (e.g., branchial arches; nose; facial mesenchyme), limb, and heart. The genomic coordinates (assembly mm10) of these elements were downloaded from the VISTA Enhancer Browser (https://enhancer.lbl.gov/)[25]. Human elements were lifted over from hg38 to mm10 via the UCSC liftOver[58] tool using *minMatch = 0.1*.

### Chromatin intersections and hidden enhancer identification
Mouse in vivo validated elements from both the VISTA Enhancer Browser and the tiling assay were intersected with tissue-specific ENCODE chromatin data via bedtools[59] (v2.29.0) to check for the presence or absence of enhancer-associated chromatin signatures (e.g., tissue-specific mouse E11.5 peaks from H3K27ac ChIP-seq, H3K4me1 ChIP-seq, and/or ATAC-seq data) within each element's genomic coordinates. We used bedtools intersect (minimum overlap of 1 bp) to evaluate the overlap (intersection) in genomic coordinates between enhancer-associated chromatin marks and the tested elements. For these intersections, we used the peak calls (BED file) generated by the ENCODE processing pipeline (see Supplementary Data 1). For mouse chromatin intersections, tested elements that were derived from human sequence were lifted over to the mouse genome (assembly mm10) via the liftOver tool (minMatch = 0.1). Similarly, for human chromatin intersections, tested elements derived from mouse sequence were lifted over to the human genome (assembly hg38) via the liftOver tool (minMatch = 0.1). We record in Supplementary Data 2

the total number of peaks (for a given chromatin dataset) that overlap with each tested element. Elements with reproducible enhancer-reporter activity (positive elements) but without any of the three enhancer-associated chromatin signatures in the relevant tissue(s) were designated as hidden enhancers. Positive elements with any (up to all) of the three enhancer-associated chromatin signatures were considered marked enhancers. Positive elements were also checked for overlap with other chromatin features available from mouse ENCODE: DNase-seq, H3K27me3 ChIP-seq, H3K36me3 ChIP-seq, H3K4me2 ChIP-seq, H3K4me3 ChIP-seq, H3K9ac ChIP-seq, and H3K9me3 ChIP-seq. Both mouse embryonic days 10.5 (E10.5) and 11.5 (E11.5) data were used for the above analyses.

## Locus selection for tiling and mouse in vivo enhancer validation

Coordinates used for the *Gli3* locus are chr13:14,626,494-15,785,614 (mm10). Coordinates used for the *Smad3/Smad6* locus are chr9:63,685,831-64,099,907 (mm10). For the tiling assay, we designed elements to span around 5000 bp each and with around several hundred bp of overlap to optimize for both coverage across the two loci (*Gli3*, *Smad3/Smad6*) and cloning efficiency. Primers were designed with flanking homology arms for Gibson cloning of the PCR amplicon into the enSERT reporter vector (Addgene plasmid #139098), which includes the mouse Shh promoter, the *LacZ* gene for enzymatic, colorimetric readout, and flanking homology arms that enable site-specific integration at the H11 locus[26]. Each tiling element was PCR amplified using mouse BACs (RPCI-23 C57BL/6 J, CHORI) as template DNA. Mean and standard deviation of the tested tiling elements: 4985 +/− 456 bp. We designed additional primer pairs for the cloning of tiling elements for which Gibson assembly initially failed. Tiling elements for which repeated cloning attempts remained unsuccessful or for which insufficient numbers of site-directed "tandem" integration transgenic embryos were obtained after multiple injections (via enSERT) were not included in the downstream analyses. Of the 350 designed tiles across the two loci, we were unable to fully test 69 (20%) elements due to incompatibility with cloning after multiple attempts with different primer pairs and/or PCR conditions (61%), or insufficient transgenic frequencies after multiple injections (39%). A mixture of the reporter construct, Cas9 protein (Integrated DNA Technologies, catalog #1081058), and sgRNAs were transferred by microinjection into the pronucleus of mouse embryos (FVB strain) and then transferred to the uterus of pseudopregnant females (CD-1 strain). Transgenic embryos were then collected at mouse embryonic day 11.5 (E11.5) for LacZ staining and the assessment of enhancer-reporter activity in several developing tissues (e.g., forebrain, midbrain, hindbrain, craniofacial, heart, and limb). For more detailed steps and information on the workflow that spans cloning, mouse colony management, microinjection, and embryo staining, refer to the recently published protocol[35]. Tested tiling elements with reproducible tissue-specific enhancer-reporter activity (as determined by a panel of researchers) in at least two independent embryos and with at least one site-specific integration were reported as positive for the given tissue. Genomic coordinates, transgenic embryo images, and tissue annotations for each element are available on the VISTA Enhancer Browser (https://enhancer.lbl.gov).

## Skeletal preparations

Mouse embryos at E18.5 were dissected in water, eviscerated, skinned and then fixed in 1 % acetic acid in EtOH for at least 24 h. Overnight staining of cartilage was performed with 1 mg/mL Alcian blue 8GX (Sigma) in 20% acetic acid in EtOH. After several washes in EtOH and treatment with 1.5% KOH for three hours, ossified tissues were stained in 0.15 mg/mL Alizarin Red S (Sigma) in 0.5% KOH for four hours, followed by de-staining in a Glycerol/KOH series[60]. Limb skeletons were imaged using a Leica MZ16 stereo-microscope coupled to a Leica DFC300Fx or DFC420 digital camera. Brightness and contrast were adjusted uniformly using Photoshop CS5.

## In situ hybridization (ISH)

Whole-mount ISH in mouse embryos was performed as previously described[61] using digoxigenin-labeled antisense riboprobes synthesized in vitro from a linearized plasmid using RNA Labeling Mix (Roche) and T3 RNA polymerase (Roche). Following fixation with 4% paraformaldehyde (PFA), embryos were washed in PBT (PBS with 0.1% Tween-20), dehydrated via a MeOH/PBT series and stored at − 20 °C in 100% MeOH. For ISH, embryos were rehydrated, bleached with 6% $H_2O_2$–PBT for 15 min and permeabilized with 10 mg ml$^{-1}$ proteinase K (PK) in PBT for 20 min. After PK treatment, embryos were incubated in 2 mg ml$^{-1}$ glycine in PBT, rinsed twice with PBT and post-fixed with 0.2% glutaraldehyde–4% PFA in PBT for 20 min. Embryos were then washed three times with PBT and transferred to prehybridization buffer (50% deionized formamide, 5 × SSC, pH 4.5, 2% Roche Blocking Reagent, 0.1% Tween-20, 0.5% CHAPS, 50 mg ml$^{-1}$ yeast RNA, 5 mM EDTA, 50 mg ml$^{-1}$heparin) for an hour at 70 °C. Subsequently, embryos were incubated overnight in hybridization buffer containing 1 mg ml$^{-1}$ Dig-labeled riboprobe at 70 °C, with gentle rotation. Post-hybridization washes were performed on the following day at 70 °C for 5 min with increasing concentrations of 2 × SSC, pH 4.5 (100% prehybridization buffer, 75% prehybridization buffer–25% 2 × SSC, 50% prehybridization buffer–50% 2 × SSC, 25% prehybridization buffer–75% 2 × SC), followed by incubation in 2 × SCC, 0.1% CHAPS twice for 30 min at 70 °C with gentle rotation. Embryos were then incubated in 20 mg ml$^{-1}$ RNase A in 2 × SSC, 0.1% CHAPS for 45 min at 37 °C, and subsequently washed in maleic acid buffer (100 mM maleic acid disodium salt hydrate, 150 mM NaCl, pH 7.5) two times for 10 min at room temperature, and two times for 30 min, at 70 °C. Samples were then extensively rinsed in TBST (140 mM NaCl, 2.7 mM KCl, 25 mM Tris-HCl, 1% Tween-20, pH 7.5), blocked with 10% lamb serum–TBST for an hour and incubated overnight at 4 °C with anti-Dig-AP antibody (Roche, 1:5,000) in 1% lamb serum. The embryos were then washed with TBST (3 ×, 5 min) to remove excess antibody, followed by five TBST washes for 1 h each, and an overnight TBST incubation at 4 °C. Embryos were then equilibrated in NTMT (100 mM NaCl, 100 mM Tris-HCl, 50 mM $MgCl_2$, 1% Tween-20, pH 9.5), followed by visualization of alkaline phosphatase activity through incubation in BM purple reagent (Roche) in the dark with gentle agitation. The reaction was terminated by five subsequent washes in PBT, each for 10 min. Treated embryos were stored long-term in 4% PFA–PBS and imaged with a Flexacam C1 camera mounted on a Leica M125C stereomicroscope.

## Evolutionary conservation

PhastCons scores were downloaded from the UCSC Genome Browser at https://hgdownload.cse.ucsc.edu/goldenPath/mm10/phastCons60way/. phastCons scores were calculated for each element (mean across region) and used to compare the levels of evolutionary conservation between different categories of tested elements (e.g., hidden enhancers vs. marked enhancers). The Kolmogorov-Smirnov test was used to assess potential differences in phastCons distributions between the considered enhancer categories.

## Additional epigenomic data

Publicly available single-cell chromatin accessibility data from mouse E11.5 forebrain (GSE100033)[38] and mouse E11.5 cerebellum (https://apps.kaessmannlab.org/mouse_cereb_atac/)[39] were used to compare differences between bulk tissue and single-cell assays in the resolution of enhancer-associated chromatin signatures, i.e., if there were open chromatin regions absent in bulk chromatin data but detected in single cell data. Human chromatin data from approximately stage-matched limb bud (GSE42413)[49], heart (GSE137731)[47], and face (GSE97752)[48] were used to evaluate if hidden enhancers from human sequence could be identified with these complementary data. The candidate cis-regulatory elements (cCRE) catalog was provided by Jill Moore and Zhiping Weng[18].

**Transcription factor motif and functional enrichment analyses**

HOMER[62] (v4.10) was used to assess, per-tissue, the enrichment of both known and de novo motifs in hidden enhancers relative to their marked counterparts, via *findMotifsGenome.pl* and the following parameters: -size given -len 8,9,10,12,14 -bg <background file = hidden and marked enhancers>. GREAT[63] v4.0.4 (http://great.stanford.edu/public/html/) was used to assess the enrichment of biological ontologies in hidden enhancers, via the basal plus extension setting (5000 bp upstream, 1000 bp downstream, distal up to 1Mbp).

**Repeat element analysis**

Repeat elements annotated across the mouse genome (by family, class, and name) were obtained from the RepeatMasker track via the UCSC Table Browser as a BED file. The number of repeat elements within each tested element's genomic interval was tallied by *bedtools intersect*[59] and used to compare the proportion of repeat element classes between enhancer mark categories (e.g., marked enhancers vs. hidden enhancers). Human elements (tested in the mouse in vivo system) were lifted over from hg38 to mm10 via the UCSC liftOver tool using minMatch = 0.1.

**Reporting summary**

Further information on research design is available in the Nature Portfolio Reporting Summary linked to this article.

## Data availability

Accession codes of previously published ATAC-seq and ChIP-seq datasets reprocessed in this study are listed in Supplementary Data 1. Wherever applicable, reference genomes Mouse GRCm38/mm10 and Human GRCh38/hg38 were used for alignment and comparisons. Images of transgenic embryos with Lacz-reporter activity are available at the Vista Enhancer Browser (http://enhancer.lbl.gov). Correspondence and requests for materials should be addressed to L.A.P. (lapennacchio@lbl.gov), A.V. (AVisel@lbl.gov) or M.O. (marco.osterwalder@unibe.ch). Source data are provided in this paper.

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

## Acknowledgements

This work was supported by U.S. National Institutes of Health (NIH) grants to L.A.P. and A.V. (UM1HG009421 and R01HG003988). M.O. was supported by the Swiss National Science Foundation (SNSF) grant PCEFP3_186993. J.L.R. was supported by grant PID2023-148267NB-I00 (MICIU/AEI/10.13039/501100011033). Research was conducted at the E.O. Lawrence Berkeley National Laboratory and performed under U.S. Department of Energy Contract DE-AC02-05CH11231, University of California (UC).

## Author contributions

M.O., D.E.D., A.V., and L.A.P. conceived the study. B.J.M. and M.O. performed critical experimental work (B.J.M., M.O.) and computational analyses (B.J.M.). B.J.M., M.O., J.A.A., S.B., E.B., T.H.G., P.G., J.G., R.D.H., E.A.L., E.M.M., Q.T.P., and K.v.M. cloned constructs for the transgenic reporter assays and performed additional experimental work related to transgenic reporter validation. B.J.M., M.O., J.A.A., and Mi.K. prepared tabular data of transgenic experiments. C.S.N., I.P.-F., Mo.K., A.N.K., S.T., and V.A. performed microinjections and surgical embryo transfers. M.O. and Y.Z. generated and characterized deletion mouse lines. I.S.-A. and J.L.-R. performed in situ hybridization studies. D.E.D., A.V., and L.A.P. provided project funding and support. B.J.M., M.O., A.V., and L.A.P. wrote the manuscript with input from the other authors.

## Competing interests

The authors declare no competing interests.
