## [Transparent Peer Review file · Nature Communications]

Uncovering Hidden Enhancers Through Unbiased In Vivo Testing

Corresponding Author: Dr Len A. Pennacchio

Version 0:

Reviewer comments:

Reviewer #1

(Remarks to the Author)

Comments:

This is an interesting study evaluating potential enhancers that do not have associated chromatin marks and thus hidden from most screening techniques. The authors do a good job of setting up the problem and retrospectively evaluate both known enhancers through the Vista enhancer browser and prospectively through a ~ sequentially mine of the loci of 2 developmentally important genes and uncover enhancers that would not be predicted by chromatin marks, i.e., hidden enhancers.

1. I find this a very interesting and novel hypothesis, and I applaud the authors on their experimental approach and design. However, I do question whether the hidden enhancers are functionally relevant in vivo, as this was not tested. I consider this a major weakness of the study. CRISPR-cas removal of a hidden enhancer that alters phenotype or survival would validate the relevance and significance of hidden enhancers. Transgenic mice showed that the isolated sequences, out of their native context, have activity, but functionality would confirm that "hidden enhancers" are more than a potential novelty of our methods for assaying activity.

2. I also question whether, if functional, the prospectively mined hidden enhancers could be species specific enhancers since the sequentially mined loci seemed to be only evaluated using murine sequences. Showing that the conserved human, or chicken sequences also retain activity would add more weight to hidden sequences being more globally important. The authors do note conservation and other commonly used genetic features of enhancers used for screening appear to be similar between hidden and marked enhancers.

3. Another point that is not clear regarding their sequential evaluation of the 2 loci is why much of the region (over ~ 27%) appeared to remain untested, i.e., grey. I did not find the explanation for why some regions went untested. Was there a consistent and unbiased rationale for skipping some sequences? This does not seem to fully capture or tile both loci and challenges the statement in the discussion that the screen represents a comprehensive and systematic assessment. The methods described an unbiased approach of 5kb fragments with overlapping tiling elements. If this was the approach used, what is the explanation for the regions that were not tested. There is likely a good explanation, I just didn't find it or missed it and I believe it is an important point that should be clearly stated for the reader.

4. A minor point, since it has been described before, yet I think it would be helpful for the authors to address how well the predicted sites (38 for Gli3 & 86 for Smad3/Smad6) identified enhancers. With only 63 of the 281 sequences showing activity to the 6 regions (88 tissue-enhancer activities), thus 65 tissue-enhancer activities out of the 124 predicted sites or about 1/2 successfully predicted activity – is that correct? Can you clarify this in the manuscript.

5. Can the authors also discuss why an enhancer element would be hidden from chromatin marks? If they are functionally active and regulating transcription, why would they be hidden from p300, Pol II etc? Are there other chromatin marks that we are missing or could these be "back-up" enhancers that only become functionally active when a similar nearby regulatory element is altered or unable to function? What is the potential relevance or significance of hidden enhancers?

Reviewer #2

(Remarks to the Author)

In this article, Mannion et al. highlight a shortcoming of current high-throughput methods of identifying enhancer sequences. By screening hundreds of sequences in the mouse genome, they show that a substantial proportion of these sequences have enhancer activity in *in vivo* assays, but do not display the histone modifications typically associated with enhancer sequences (e.g., H3K27ac, H3K4me1). Despite their lack of overlap with many traditional annotations, these enhancer sequences do display other features of enhancers, such as increased evolutionary conservations, and are similar to their marked counterparts in sequence features, such as transcription factor binding sites and transposable element composition.

Major comments:

1. Did the tissue-enhancer pairs without chromatin marks (discussed in Figure 3) from different tissues overlap any of the same genomic regions? Do the authors suspect a biological reason for why some tissues have a higher proportion of tissue-specific hidden enhancers, and were there difference across tissues? This is not completely clear in the results or discussion.
2. There is correlation between enhancer activity and expression of the Gli3 or Smad3/ Smad6 genes, but is there external evidence that the tested enhancers are targeting the Gli3 or Smad3/ Smad6 genes? This work seems to show that there are many potential enhancer sequences; it would be helpful to have a stronger line of logic connecting the activity of one or more of these enhancers to the gene expression.
3. Previous work has shown some evidence for enhancer-promoter specificity in *Drosophila* (e.g., Zabidi et al. 2015) or cooperation between enhancers leading to the observed expression output (e.g., Bergman et al. 2022). Could any of the hidden enhancers provide evidence for this in mammalian species? Are there differences in the number of hidden enhancers observed between the retrospective VISTA analysis and the site-directed approach?
4. The GREAT analysis is dependent on proximity to genes for the downstream analysis. Since the enhancers were taken the loci (one of which is in a known gene desert), I am skeptical that we can interpret this analysis. It would be helpful to know which genes were selected for the enrichment analysis. No data is provided in Table S6.
5. The chromatin/sequence attributes of the tested enhancer rely on the calculation of overlap between the enhancer location and the locations of ChIP-seq peaks or other BED files. However, the methods do not adequately describe how intersections were calculated. Are all intersections considered binary? The authors should clearly describe the parameters used for the bedtools intersect command in the Methods.
6. On a similar note, the tested regions are relatively long (5kb). I am concerned that the length of the sequence may influence the number of potential intersections, make it difficult to compare between histone modification ChIP-seq intersections and other attributes, such as PhastCons elements. For annotations that are systematically shorter than the enhancer sequences, how is length accounted for? How much noise is introduced by having enhancer “regions” that may be longer than the functional part of the sequence?
7. Broadly, the Methods sections are quite sparse in detail. They would benefit from additional detail and/or references to clearly outline the experimental and computational approaches. In addition to the items in 4-5, how was 5kb selected? Why are some of the regions not validated? How long are the sequence constructs in the assay?

Minor comments:

1. Figure 1d is missing the proportion label for the 3+ marks category (shown in gray).
2. Figure 3a is also missing the proportion label for 3+ marks category (shown in gray).
3. It is difficult to distinguish the titles for each set of alternate data from the subtitles in Figure 4c. Consider making the titles a larger font or highlighting the schematic (a-c) along with the relevant title.
4. The color scales in the Figure S5 and S6 heatmaps are difficult to parse when expression is high (especially Fig S6c). They would benefit from numeric annotations.

Reviewer #3

(Remarks to the Author)

In this study, the authors carry out a tiling transgenic approach in two mouse loci to systematic assay for enhancer activity, and then relate findings to ENCODE chromatin signatures in concordant tissues. They find that “14% of enhancers did not show canonical enhancer-associated chromatin signatures in the tissue in which they are active”. This is an interesting study that demonstrates the limitations of current epigenomic annotations.

Perhaps the most important clarification to be made about this study is that it does not show that some enhancers do not actually have these chemical modifications, instead they have been unable to capture them in existing datasets – largely ENCODE and published experiments with E11.5 forebrain and cerebellum scATAC. The fact that they cannot be detected is important, and the most plausible explanations can be inferred from the studies they perform. For example, the modifications could be present in a small proportion of cells in dissected tissue, so that the signal to noise is insufficient to enable detection. This is consistent with several examples where the enhancer Lacz activity is clearly restricted to very few cells (e.g. Figure S6). Another plausible explanation is that ChiP-seq/ATAC data is not of sufficient quality, typically because of well known challenges to obtain high quality data from dissected mouse embryonic tissue. In fact the Chip-seq data shown in figures 2 and 3 shows considerable background (which is quite usual in data that is obtained from ultra small tissue). In the case of scATAC, it is unclear if the coverage is truly exhaustive for less prevalent cell types, or how many accessible sites might not show signal above background yet still lacking statistical significance. The authors carefully examined earlier (E10.5) timepoints to rule out residual Lacz activity from a preceding stage, but the same caveats apply here: those assays in E10.5 tissues can miss activity from few cells, or because of the same type of technical limitations outlined above. Finally,

the authors find that the vast majority of the enhancers that “lack” active enhancer chromatin signatures actually do have enhancer activity marks in another tissue (although that is also seen for inactive DNA elements, so in the end there is only some enrichment, unclear if significant, in active elements).

Every one of the analyses they came up with effectively reduced the burden of hidden examples, which suggests that the residual number of hidden enhancers would be further reduced if they had access to more types of datasets: much larger and robust single cell datasets, ChIP-seqs with more pure tissues, perhaps even robust DNA methylation data, etc. A significant message of this study, therefore, is that it highlights the limitations of mouse ENCODE data, and calls for the generation of more extensive high quality datasets.

In conclusion, the authors state that they have identified “enhancers that are hidden from proper tissue-specific genome annotation using current chromatin-based enhancer identification approaches”, and “a substantial proportion of in vivo enhancers are hidden from discovery by conventional chromatin profiling methods”. But it this seems to be more accurately “using current chromatin-based enhancer datasets” rather than the approaches or methods themselves. It is very likely that more exhaustive, higher quality datasets using the same methods do reveal active enhancer signatures.

This distinction is interesting because previous studies have suggested that some enhancers have non-canonical modified histone marks. Furthermore, whether some enhancers might intrinsically not exhibit accessible chromatin in current assays (ATAC, DNase1). All of this is intriguing but is untested by the current study. The abstract, therefore, needs to be adapted to fully clarify that the reason why some enhancers are not detected is a limitation of current datasets, rather than the methods per se.

Another major point is that in the end the fraction of “active enhancers” that lacks marks is not so high. An alternative explanation is that they are false positives of this artificial enhancer assay, rather than false negatives of the chromatin marks. Unless the authors feel that this conundrum can be resolved, they may wish to incorporate this possibility to the conclusion.

Other specific points

- As outlined above, some examples might be false positives rate of this artificial assay, which does not take into account how enhancers might actually work in their native genomic sites. While this reviewer cannot think of a simple gold standard method to benchmark the method, much more information on replication of enhancer assays concerning expression patterns is essential to assess these findings. This should be clear in the main text and methods. So far the information that is provided is largely qualitative, how consistent are the expression patterns. Can authors provide more compelling data than just showing a handful of examples?
- The overlaps with other tissues is interesting, but given the “background overlap” it would be more informative to ask if there are overlaps with snATAC specifically from other neural lineage clusters, where the priors for overlaps being meaningful are higher, and with more strict spatial resolution criteria.
- The authors have studied mainly mouse ENCODE, and two scATAC datasets. Have they considered using other datasets? If no other valid dataset exists this should be explained.
- Discussion: “we could not identify sequence, genomic, or other epigenomic properties that could distinguish hidden enhancers from their marked counterparts”. Please explain this sentence, it seems to me that the authors do claim that there are epigenomic differences.

Reviewer #4

(Remarks to the Author)

Comments on the manuscript:

Uncovering Hidden Enhancers Through Unbiased In Vivo Testing

In this manuscript, Mannion, Visel, Pennacchio, and collaborators combine in vivo functional genomic approaches with ChIP-seq and ATAC-seq data to interrogate the correlation between identification of enhancers using classic chromatin features (i.e. H3K4me1, H3K27ac, and ATAC-seq) versus functional identification using in vivo transgenic enhancer reporter assays.

The authors conclude that using H3K4me1, H3K27ac, and ATAC-seq is insufficient for a comprehensive annotation of enhancers in the mouse genome. To prove this, they compare enhancer identification by combining stage and tissue-matched H3K4me1 and H3K27ac ChIP-seqs and ATAC-seq with identification based on in vivo reporter assays. Some of these in vivo reporter assays were already present in their VISTA Enhancer Database and others have been performed to address the hypotheses in this manuscript.

A preliminary “retrospective” search for classic chromatin features on functionally validated enhancers from the VISTA database indicated that hidden enhancers comprise from 9 to 25% of all the VISTA database enhancers. This observation motivated the authors to analyze further if the existence of hidden enhancers is a widespread feature of the mouse genome. To assess this, they divided the Gli3 and Smad3/Smad6 genomic landscapes into 5 kb bins, clone these DNA regions into reporter plasmids and performed transgenic in vivo reporter assays. The authors observed that 26% of the ~5kb tiles that were positive in the in vivo functional assays lacked enhancer classical marks in stage and tissue-matched NGS data. Then, in figure 3, the authors utilize non-classical enhancer histone marks to evaluate if hidden enhancers display any of these features. Their data shows that hidden enhancers do not contain any of these features (H3K27me3, H3K36me3, H3K4me2, H3K4me3, H3K9ac/H3K9me3). Next they compare other enhancer features such as evolutionary conservation,

transcription factor binding sites, and presence of transposable elements and they conclude that none of these features distinguishes “canonical” from “hidden” enhancers.

Finally, the authors consider alternative hypotheses that could explain the lack of classic enhancer chromatin features in hidden enhancers. Their results conclude that some of the hidden enhancers at E11.5 displayed classical enhancer features at E10.5, thus the positive result in the transgenic reporter assays is likely due to residual lacZ reporter activity from activation in an earlier developmental stage. They also draw upon available snATAC-seq to test if tissue-heterogeneity contributes to the lack of chromatin accessibility in the bulk experiments and use human epigenomic data of syntenic regions to try to predict those hidden enhancers. Interestingly, the authors report that ENCODE cis-regulatory elements catalog that represents multiple cell lines and tissues does identify ~90% of the hidden enhancers.

Overall, the results presented in this manuscript confirm observations made by many in the field. Using classical marks is a good approximation for enhancer identification, but their comprehensive annotation in a cell and tissue-specific manner requires functional assessment in the precise chromatin environment (i.e. CRISPR-Cas9 deletion, dCas9-KRAB repression, reporter assays).

From my point of view, this manuscript lacks the scientific novelty expected from a Nature Communications research article. The existence of hidden enhancers is not sufficiently supported by the data. For this reviewer, the data only supports the lack of accurate measurements of chromatin features in cell types that may be rare in nature and reflects the exquisite spatial and temporal control of gene expression by enhancers. Alternatively, most of their lacZ reporter positive signals may be the product of an earlier developmental activation of the reporter, and this could explain the lack of correlation with the chromatin features. Last, selection of H3K4me1, H3K27ac, and accessibility as main chromatin features seems rather outdated. Nonetheless, this reviewer would like to remark that the functional assessment of the tiled Gli3 and Smad3/Smad6 genomic landscapes by in vivo reporter assays is an elegant approach and as they have included the results in the VISTA database, it will be a useful resource for scientists in the field.

Major comments:

1-One of the potential explanations for the lack of classic histone modifications and accessibility at hidden enhancers could be tissue heterogeneity. H3K27ac, H3K4me1, and ATAC-seq assays were performed in bulk tissue, and thus they reflect enrichment across the population of cells. If a cell-type is scarce, it could still test positive in transgenic assay while the overall background would hinder the enrichment by ChIP-seq or ATAC-seq. In my opinion, it is not sufficiently proven that these enhancers do not exhibit chromatin marks, thus limiting the interpretation of the main conclusions of this manuscript (i.e. classic histone marks and accessibility are not enough for identifying enhancers).

2-The authors partially address major comment 1 by analyzing snATAC-seq of forebrain and hindbrain in Figure 4B, and they report enrichment of ATAC-seq signal in 8 enhancers, 2% of the total hidden enhancers. This percentage is misleading and it should refer to the total of hidden forebrain and hindbrain enhancers (n=44). With this calculation 18% of forebrain and hindbrain hidden enhancers display classic enhancer ATAC-seq signal when evaluated in a single cell setting.

Minor comments:

1-The Vista Database candidate enhancers were selected by means of p300 enrichment or evolutionary conservation. It would be useful to know how many of the hidden enhancers were in fact originally identified by using p300 ChIP-seq.

2-In my opinion, the nomenclature “hidden” enhancers is misleading, as it invokes the idea of a novel class of enhancers. The authors use this word to distinguish hidden enhancers from enhancers marked by classical enhancer features, but for this reviewer, the lack of chromatin marks in the hidden enhancers has not been sufficiently proven in this manuscript. Additionally, one could argue that their vision of classical marks is outdated. Researchers these days combine H3K4me1/H3K27ac/ATAC-seq, transcription factor binding sites clusters, RNAPII presence, 3D-chromatin measurements etc. to identify enhancers.

3-Figure 1c. is missing the scale of the signal.

4-Authors should indicate what is the meaning of the numbers in the chromatin features columns of Supplementary Table 2.

5-Specific command used for intersecting chromatin features and enhancers using bedtools should be detailed, as too stringent colocalization requirements may underestimate the number of canonical enhancers.

6-Authors conclude that 56% of the initial hidden enhancers could in fact be identified using chromatin features from E10.5, suggesting that the lacZ activity is residual from an earlier activation. This reviewer positively appreciates this analysis and suggests that more details about the pitfalls of transgenic in vivo assays could be provided in the discussion.

7-To rule out the possibility of hidden enhancers being marked by “non-classical” histone marks, the authors use ENCODE data of K27me3, K36me3, K4me2/3, K9ac, and K9me3. Results in Figure 3b and Supplementary Figure 7 conclude none of these marks are present at the hidden enhancers. This is not explained in the text at all.

8-In the search for differences between canonical and hidden enhancers, the authors compare enriched TFBSs concluding that there are not particular motifs enriched. As it is, this analysis is not meaningful unless the hypothesis is that hidden enhancers are a novel class of enhancers regulated by a common set of TFs across tissues. In my opinion, this analysis would bring more utility when performed in a tissue-by-tissue fashion, where presence or absence of a motif in a hidden enhancer could provide relevant biological information (for example reveal a temporal aspect of the activation of the enhancer that could hint reasons why there is a disconnection between chromatin features and in vivo activity).

Version 1:

Reviewer comments:

Reviewer #1

(Remarks to the Author)

The authors have responded thoroughly to the reviewers questions, suggestions, and critiques.

This is an important body of work that will benefit those searching for tissue-specific enhancers. I am pleased to see that they were able to correlate function with transgenic activity of two hidden enhancers. This is a significant amount of work, and I believe their diligence has made this an even more important and citable manuscript. I realize that redundancy can obscure the functional contributions of enhancers and I applaud the authors for navigating through this challenge using a compromised genetic (Gli3^d) background to accentuate the contribution of these hidden enhancers in anterior limb patterning. Excellent work!!

I do have one critique/recommendation. The authors describe a CRISPR/cas9 deletion upstream of Gli3. I think this deletion is named iTAD by the authors but it is not defined in the results section. When the abbreviation shows up later in the paragraph – it is again not defined. I would ask the authors to define iTAD for the readers, probably naming the deletion when it is first described in the results. Later in the discussion this deletion is again described but not named. “deletion of a nearly 300kb Gli3-upstream genomic interval containing multiple canonical and designated hidden enhancer elements” It would benefit to the reader to have the name again attributed to this deletion by adding in parentheses iTAD (iTAD).

Looking forward to referencing this paper in the future!
All the best, Kerby Oberg

Reviewer #2

(Remarks to the Author)

I thank the authors for their clear and complete responses to the reviews. I believe the manuscript is much clearer now and can be accepted. I only have 2 minor comments:

- There is a typo in H3K27ac in line 381 (Discussion).

- In Line 394 the phrase: “...we can expect to better understand and the unique characteristics of hidden enhancers...”.

However, in the paper you do not find strong evidence that these sequences are functionally different from marked enhancers (outside of their identification strategy). This sentence could be clarified if you do not think that hidden enhancers are a separate class of element (which seems to be the case given the rest of the paper and response to reviewers).

Reviewer #4

(Remarks to the Author)

The authors have addressed all my concerns. I appreciate the nuance regarding the technical limitations of detecting the "hidden" enhancers and the clarifications regarding their "technical" versus "biological" nature.

I support the publication of the article as it is.

Congratulations to the team and thank you for your valuable resources for the enhancer field!

made.

Summary Response to Reviewers Comments

NCOMMS-23-00294-T: Uncovering Hidden Enhancers Through Unbiased In Vivo Testing

We thank the reviewers for their comments. Below we have summarized our responses to the points raised and changes to the manuscript.

Reviewer 1

Reviewer #1 Remarks to the Author

This is an interesting study evaluating potential enhancers that do not have associated chromatin marks and thus hidden from most screening techniques. The authors do a good job of setting up the problem and retrospectively evaluate both known enhancers through the Vista enhancer browser and prospectively through a ~ sequentially mine of the loci of 2 developmentally important genes and uncover enhancers that would not be predicted by chromatin marks, i.e., hidden enhancers.

We thank the reviewer for their appreciation of the scale of this study, its conceptual advance in demonstrating that current chromatin-based screening techniques miss many bona fide enhancers, and the detailed characterization of the enhancer landscape at two major developmental loci through extensive mouse experiments.

1. I find this a very interesting and novel hypothesis, and I applaud the authors on their experimental approach and design. However, I do question whether the hidden enhancers are functionally relevant *in vivo*, as this was not tested. I consider this a major weakness of the study. CRISPR-cas removal of a hidden enhancer that alters phenotype or survival would validate the relevance and significance of hidden enhancers. Transgenic mice showed that the isolated sequences, out of their native context, have activity, but functionality would confirm that "hidden enhancers" are more than a potential novelty of our methods for assaying activity.

We appreciate the thoughtful suggestion by this reviewer. We agree that demonstrating enhancer function through knockout experiments can provide further support for the *in vivo* function of a given enhancer.

To address this question, we performed additional extensive experimentation using CRISPR/Cas9 engineering and mouse genetics to interrogate the function of hidden enhancers at the *Gli3* locus, one of the two loci that we systematically screened using our tiling reporter transgenesis approach (described in Fig. 2). As shown now in the new Supplementary Figure 8 (Fig. S8A, B), we deleted a 303kb intra-TAD (iTAD^d) region upstream of *Gli3*, containing four hidden and seven canonical enhancers. While our previous study¹ as well as the work of other labs has demonstrated that developmental enhancer function is frequently masked by compensating enhancer activities or

redundant effects conferring robustness at the transcriptional level^{1,2}, we reasoned that this experimental setup would specifically allow us to investigate the function of a particular hidden limb enhancer: mm2164. Mm2164 drives anterior limb activity which overlaps with the activity of the nearby mm1179 canonical limb enhancer identified and analyzed in our previous studies^{1,3}. Mm1179 is also located within the deleted interval (iTAD^d; Figure S8A). Remarkably, CRISPR/Cas9-mediated engineering of mice lacking the 303kb iTAD interval (Fig. S8B) did not reveal overt phenotypes and led to viable mice, including normal limb morphology at E18.5 (Fig. S8C). As mm2164 and mm1179 represent the only known limb enhancers identified within the deleted interval, we decided to investigate mm2164 hidden enhancer function in the presence of reduced target gene dosage and took advantage of an available *Gli3* knockout allele¹. Using such a “sensitized” strategy analogous to our previous approach on the mm1179 enhancer¹ allowed us to reveal a functional requirement for the mm2164 hidden enhancer (Fig. S8D). We previously demonstrated that mm1179^d/*Gli3*^d limbs exhibit a duplicated digit 1 and a normal digit 2 with full penetrance in the LFL, but with n=4/7 embryos showing variable bi-furcation of digit 2¹ (Fig. S8D). However, in absence of both mm1179 and mm2164 limb enhancers (as part of the iTAD deletion) in *Gli3* sensitized background (iTAD^d/*Gli3*^d), we observed an exacerbated digit phenotype, marked by duplication of digit 1 with additional bifurcation of digit 2 in both the LFL and the RFL (Fig. S8D). Therefore, these results support that the hidden enhancer mm2164 (likely during early limb development) is required at the functional level for limb development by contributing to robust *Gli3* dosage in the anterior limb mesenchyme known to control anterior digit number⁴.

We now describe the 303kb deletion within the large non-coding *Gli3* upstream interval and discuss the consideration described above as follows:

Results: Additionally, using CRISPR/Cas9 genome editing, we generated a mouse line harboring a deletion of a 303kb genomic interval upstream of the *Gli3* TSS. The deleted region encompasses seven canonical and four hidden enhancers, each with distinct tissue-specific activities (Fig. S8A). Remarkably, homozygous deletion of this interval did not reveal any gross phenotypic changes and limb morphology remained normal despite the absence of the mm1179 (canonical)³ and mm2164 (hidden) limb enhancers (Fig. S8B, S8C). While this result was in line with our prior study revealing compensation of individual (mm1179) enhancer function through redundant interactions¹, we leveraged our *Gli3* deletion allele¹ to investigate the functional contribution of the mm2164 hidden enhancer in direct comparison with mm1179 at reduced *Gli3* dosage (Fig. S8C, S8D). For this purpose, we compared limb skeletons of *Gli3*^{d/+}, mm1179^d/*Gli3*^d and iTAD^d/*Gli3*^d embryos. A split digit 1 phenotype observed in *Gli3*^{d/+} embryos is exacerbated in mm1179^d/*Gli3*^d embryos, which show digit 1 duplication¹. In addition, a subset of mm1179^d/*Gli3*^d embryos (n=4/7) also exhibited bi-furcation of digit 2, but only in the right forelimb

(RFL)¹. Upon iTAD deletion (iTAD^d/Gli3^d), which removes both the previously known limb enhancer mm1179 and the newly discovered enhancer mm2164, we observed a slightly more compromised limb phenotype than in mm1179^d/Gli3^d embryos (**Fig. S8D**). Specifically, in addition to duplication of digit 1, we observed a substantial increase in the frequency of additional bifurcation of digit 2 in iTAD^d/Gli3^d embryos that affected predominately the left forelimb (LFL, Fig. S8D). In iTAD^d/Gli3^d embryos, we observed LFL digit 2 bifurcation in 2 of 3 embryos (67%), compared to 0 of 7 mm1179^d/Gli3^d embryos (0%). The observed left-right asymmetry of the phenotype is consistent with known asymmetries in limb development and genetic syndromes affecting predominately the left or right limb⁵. This result supports that the hidden enhancer mm2164 is functional and likely contributes to robustness of early anterior *Gli3* gene dosage required for anterior digit control^{1,4}.

Discussion: Remarkably, the deletion of a nearly 300kb *Gli3*-upstream genomic interval containing multiple canonical and designated hidden enhancer elements with tissue-specific activities did not result in overt phenotypic alterations, indicating the presence of multiple tissue-specific redundant enhancers outside of the deleted region. This is similar to prior observations at the *Gli3* locus and highlights the challenges in functional enhancer dissection when multiple enhancers can provide redundancy or a buffering effect to maintain gene expression¹. However, using our previously applied genetic framework for evaluation of *Gli3* enhancer function in a sensitized background¹, we find evidence for the functionality of hidden enhancers. As observed for developmental enhancers of other TF genes, the function of a hidden limb enhancer upstream of *Gli3* (mm2164) appears to contribute to phenotypic robustness^{1,6}, yet future work to demonstrate roles and potential differences of both marked and designated hidden enhancers in other loci will be required.

2. I also question whether, if functional, the prospectively mined hidden enhancers could be species specific enhancers since the sequentially mined loci seemed to be only evaluated using murine sequences. Showing that the conserved human, or chicken sequences also retain activity would add more weight to hidden sequences being more globally important. The authors do note conservation and other commonly used genetic features of enhancers used for screening appear to be similar between hidden and marked enhancers.

We thank the reviewer for this interesting thought. As outlined in the main text, we found that “both hidden and marked enhancers have similar levels of evolutionary conservation, *i.e.*, both categories have elevated conservation scores (phastCons) relative to genomic

background and show no significant difference in the level of conservation (Fig. S9)". Taking into account (canonical) human-mouse conserved Vista enhancers for which both mouse and human versions were tested in transgenic reporter assays, we observe conservation of reporter activities in 4/8 cases (in tissues defining hidden enhancer identity - see below). Therefore, based on the similar level of average phastCons score relative to genomic background when comparing canonical and hidden enhancers, we conclude that human and mouse hidden enhancers drive similarly conserved activities. Determining whether hidden enhancers are biased towards "species specificity" (e.g., by testing human and/or chicken elements) in general would require considerable resources due to the substantial number of additional enhancer elements to be tested in mouse embryos and would lead to a significant additional delay of the timeline for these revisions.

Table of tissue-specific hidden enhancers with conserved activities in mouse and human transgenic reporter assays (data available through our **Vista Enhancer browser**: <https://enhancer.lbl.gov/vista/>):

Human Ortholog		Mouse Ortholog	
hs638	hidden Enh (limb) at E11.5	mm2107	positive in limb at E11.5
hs1112	hidden Enh (craniofacial) at E11.5	mm1643	positive in craniofacial tissue at E12.5
hs1937	hidden Enh (craniofacial, hindbrain) at E11.5	mm1823	positive in craniofacial tissue at E11.5
hs2731	positive in limb at E13.5	mm2022	hidden Enh (limb) at E11.5

3. Another point that is not clear regarding their sequential evaluation of the 2 loci is why much of the region (over ~ 27%) appeared to remain untested, i.e., grey. I did not find the explanation for why some regions went untested. Was there a consistent and unbiased rationale for skipping some sequences? This does not seem to fully capture or tile both loci and challenges the statement in the discussion that the screen represents a comprehensive and systematic assessment. The methods described an unbiased approach of 5kb fragments with overlapping tiling elements. If this was the approach used, what is the explanation for the regions that were not tested. There is likely a good explanation, I just didn't find it or missed it and I believe it is an important point that should be clearly stated for the reader.

We attempted to clone all 350 designed tiles (~5 kb) across the two loci. Of these 350, we cloned and fully tested 281 (80%). The remaining 20% were either 1) unable to be

cloned after multiple attempts with different primer pairs and/or PCR conditions or 2) did not yield sufficient transgenics to assess for reproducible enhancer-reporter activity at the conclusion of this study. We have added the following to the methods section:

Of the 350 designed tiles across the two loci, we were unable to fully test 69 (20%) elements due to incompatibility with cloning after multiple attempts with different primer pairs and/or PCR conditions (61%), or insufficient transgenic frequencies after multiple injections (39%).

4. A minor point, since it has been described before, yet I think it would be helpful for the authors to address how well the predicted sites (38 for Gli3 & 86 for Smad3/Smad6) identified enhancers. With only 63 of the 281 sequences showing activity to the 6 regions (88 tissue-enhancer activities), thus 65 tissue-enhancer activities out of the 124 predicted sites or about ½ successfully predicted activity – is that correct? Can you clarify this in the manuscript.

We have added the following to the Results section:

Of those regions across the two loci that were predicted to be enhancers based on tissue H3K27ac data and that were subsequently tested for enhancer reporter activity, we found the validation rate of H3K27ac across the six tissues considered to be 28% (**Table S4**). We found that regions with high ranked H3K27ac signal (*i.e.*, based on H3K27ac signal enrichment) corresponded with higher validation rates than those tested regions with low ranked H3K27ac signal, which is consistent with previous studies.

5. Can the authors also discuss why an enhancer element would be hidden from chromatin marks? If they are functionally active and regulating transcription, why would they be hidden from p300, Pol II etc? Are there other chromatin marks that we are missing or could these be "back-up" enhancers that only become functionally active when a similar nearby regulatory element is altered or unable to function? What is the potential relevance or significance of hidden enhancers?

Please also see our responses to Reviewer 2, Point 2 and Reviewer 3, Point 2.

We have now expanded our Discussion to consider these suggestions as follows:

Nonetheless, we cannot yet resolve whether the observation of these hidden enhancers is attributable to a novel biological mechanism or instead to technical limitations of the current data. Of the hidden enhancers identified in this study, it is likely that a majority do have enhancer-associated chromatin marks which were missed due to the presence of these marks in a limited number of cells or during a time point not captured by these data. We were unable to find any distinguishing properties of hidden enhancers using additional histone modifications profiled in the ENCODE developmental chromatin catalog. In this study we could not systematically exclude in a tissue- and developmental stage-matched manner the possibility that other histone

modifications^{7,8} (e.g., H3K277ac; H2BNTac) or proteins⁹⁻¹¹ (e.g., p300/CBP, Pol II) are present at these enhancers. In consideration of these technical limitations and also their indistinguishability from marked enhancers based on sequence conservation and other genomic properties, we suggest hidden enhancers to be functionally relevant contributors to the regulatory landscapes in which they are present.

Reviewer 2

Reviewer #2 Remarks to the Author

In this article, Mannion et al. highlight a shortcoming of current high-throughput methods of identifying enhancer sequences. By screening hundreds of sequences in the mouse genome, they show that a substantial proportion of these sequences have enhancer activity in in vivo assays, but do not display the histone modifications typically associated with enhancer sequences (e.g., H3K27ac, H3K4me1). Despite their lack of overlap with many traditional annotations, these enhancer sequences do display other features of enhancers, such as increased evolutionary conservations, and are similar to their marked counterparts in sequence features, such as transcription factor binding sites and transposable element composition.

We thank the reviewer for acknowledging the scale of the present study and their appreciation for our key finding that traditional chromatin-based methods routinely “miss” bona fide enhancers in the genome.

1. Did the tissue-enhancer pairs without chromatin marks (discussed in Figure 3) from different tissues overlap any of the same genomic regions? Do the authors suspect a biological reason for why some tissues have a higher proportion of tissue-specific hidden enhancers, and were there difference across tissues? This is not completely clear in the results or discussion.

We thank the reviewer for this interesting question. For certain tissues we observe a higher level of concordance in terms of enhancer-reporter activity from the same genomic region (see Figure 1 for Reviewers below). In particular, we observe a higher degree of tissue-overlapping reporter activities in forebrain, midbrain, and hindbrain (all of which are nervous system tissues) for both marked and hidden enhancers. However, we do not detect a significant difference in tissue-specificity patterns when comparing marked and hidden enhancers, but have now added the following description to the discussion to better inform about this point:

Similarly, forebrain, midbrain, and hindbrain activities were detected most frequently among both hidden and marked enhancer categories, likely related to the complexity of the regulation of brain development¹².

Figure 1 for Reviewers: Correlation plot showing concordance among forebrain/midbrain/hindbrain and craniofacial/limb elements for marked and hidden enhancers.

2. There is correlation between enhancer activity and expression of the *Gli3* or *Smad3/Smad6* genes, but is there external evidence that the tested enhancers are targeting the *Gli3* or *Smad3/Smad6* genes? This work seems to show that there are many potential enhancer sequences; it would be helpful to have a stronger line of logic connecting the activity of one or more of these enhancers to the gene expression.

In absence of high-resolution Hi-C datasets across a diverse range of mouse embryonic tissues and capturing of interactions at *Gli3* or *Smad3/6* loci (e.g., via Capture-HiC), it remains challenging to delineate the exact promoter specificities of marked/hidden enhancers at *Gli3* and *Smad3/6* loci. In addition, tissue-specific Hi-C and Capture-HiC datasets, even if at considerable depth, are frequently not sufficient to unequivocally determine the kinetics and targets of enhancers with weaker activities or those only active in smaller cell subsets¹³. However, given that *Gli3* and *Smad3/6* loci represent the only protein-coding genes with pronounced developmental function and expression patterns within in their respective TADs^{14,15}, our previous studies suggest that overlap between enhancer activity patterns (determined by transgenic reporter analysis) and target gene expression can be considered strong evidence for functional enhancer-target gene interaction^{1,2,16}. We also interrogated our VISTA enhancers against a recent Capture-HiC dataset¹⁶, which revealed 3D-interactions of nearly n=40 individual hidden enhancer elements with canonical Vista enhancers (baits) across tissues, now listed in Table S7. This further corroborates functional enhancer characteristics of at least a subset of hidden enhancer elements.

Please also see our response to Reviewer 1, Point 5.

3. Previous work has shown some evidence for enhancer-promoter specificity in *Drosophila* (e.g., Zabidi et al. 2015) or cooperation between enhancers leading to the observed expression output (e.g., Bergman et al. 2022). Could any of the hidden enhancers provide evidence for this in mammalian species? Are there differences in the number of hidden enhancers observed between the retrospective VISTA analysis and the site-directed approach?

Evaluating enhancer-promoter specificity (e.g., Zabidi et al. 2015) or potential multiplicative effects between enhancers and promoter classes (e.g., Bergman et al. 2022) are outside the scope of our implemented enSERT (site-directed transgenic reporter) strategy to accurately identify tissue-specific and target gene-overlapping activities in mouse embryos. The retrospective VISTA analysis is based on a set of highly reproducible enhancers from previous studies with strong tissue-specific activities, a subset of which has demonstrated to interact with presumed target genes based on enhancer activity-gene expression correlation¹⁶. Our observations that hidden enhancers generally show tissue-specific reporter activities similar to their marked counterparts (enhancers with canonical signatures) and correlate with target (*Gli3*) gene expression (Fig. S7), suggest nevertheless that at least hidden enhancers with strong activities are likely to show similar enhancer-promoter interaction specificities as those analyzed previously¹⁶. Evaluating the recently published Capture-C dataset targeting hundreds of VISTA enhancers¹⁶ revealed interactions of hidden enhancers with elements contacted by canonical enhancers (n=262 interactions), further indicating dynamic and functional properties to hidden enhancers (Table S7). Based on the frequency of hidden enhancers identified through enSERT across two loci in our study, we expect a considerable number of hidden enhancers to be present throughout the genome and especially near other developmental loci.

Please also see our response to Reviewer 1, Point 1.

4. The GREAT analysis is dependent on proximity to genes for the downstream analysis. Since the enhancers were taken the loci (one of which is in a known gene desert), I am skeptical that we can interpret this analysis. It would be helpful to know which genes were selected for the enrichment analysis. No data is provided in Table S6.

In our original GREAT analysis, we used both retrospective and tiling studies to compare hidden enhancers with marked enhancers on a tissue-by-tissue basis (e.g., hidden forebrain enhancers vs. marked forebrain enhancers). We agree that such a comparison would be affected by the cluster of enhancers identified across the two loci from the tiling study and we apologize for the lack of clarity regarding this analysis. We have therefore redone the GREAT (GREAT v4.0.4) analysis as follows:

From the retrospective VISTA study only, we checked for potential differences between hidden and marked enhancers for each of the six tissues by only supplying coordinates

of genomic enhancer regions (e.g., hidden limb enhancers vs. marked limb enhancers). For 5 of the 6 tissues (forebrain, midbrain, hindbrain, craniofacial structure, limb), no differences in gene ontology processes or mouse phenotypes were observed between hidden and marked enhancers.

5. The chromatin/sequence attributes of the tested enhancer rely on the calculation of overlap between the enhancer location and the locations of ChIP-seq peaks or other BED files. However, the methods do not adequately describe how intersections were calculated. Are all intersections considered binary? The authors should clearly describe the parameters used for the bedtools intersect command in the Methods.

We have now clarified the methodology as follows in the Methods section:

We used bedtools intersect (minimum overlap of 1bp) to evaluate the overlap (intersection) in genomic coordinates between enhancer-associated chromatin marks and the tested elements. For these intersections, we used the peak calls (BED file) generated by the ENCODE processing pipeline (see **Table S1**). For mouse chromatin intersections, tested elements that were derived from human sequence were lifted over to the mouse genome (assembly mm10) via the liftOver tool (minMatch=0.1). Similarly, for human chromatin intersections, tested elements derived from mouse sequence were lifted over to the human genome (assembly hg38) via the liftOver tool (minMatch=0.1).

6. On a similar note, the tested regions are relatively long (5kb). I am concerned that the length of the sequence may influence the number of potential intersections, make it difficult to compare between histone modification ChIP-seq intersections and other attributes, such as PhastCons elements. For annotations that are systematically shorter than the enhancer sequences, how is length accounted for? How much noise is introduced by having enhancer “regions” that may be longer than the functional part of the sequence?

For evolutionary conservation, we compared the mean phastCons scores (only using regions that did have a score, the “mean covered only” option) for all elements. For chromatin-based data (i.e., ChIP-seq, ATAC-seq), we did not account for the lengths of annotated regions since intersections between enhancer sequence and an annotated region were only evaluated in a binary case (i.e., a tested enhancer sequence either has 1 or more chromatin annotated regions or does not).

7. Broadly, the Methods sections are quite sparse in detail. They would benefit from additional detail and/or references to clearly outline the experimental and computational

approaches. In addition to the items in 4-5, how was 5kb selected? Why are some of the regions not validated? How long are the sequence constructs in the assay?

We have now included the following in the Methods section:

For the tiling assay, we designed elements to span around 5,000bp each and with around several hundred bp of overlap to optimize for both coverage across the two loci (*Gli3*, *Smad3/Smad6*) and cloning efficiency. Primers were designed with flanking homology arms for Gibson cloning of the PCR amplicon into the enSERT reporter vector (Addgene plasmid #139098), which includes the mouse *Shh* promoter, the *LacZ* gene for enzymatic, colorimetric readout, and flanking homology arms that enable site-specific integration at the H11 locus¹⁷. Each tiling element was PCR amplified using mouse BACs (RPCI-23 C57BL/6J, CHORI) as template DNA. Mean and standard deviation of the tested tiling elements: 4985 +/- 456bp. We designed additional primer pairs for cloning of tiling elements for which Gibson assembly initially failed. Tiling elements for which repeated cloning attempts remained unsuccessful or for which insufficient numbers of site-directed "tandem" integration transgenic embryos were obtained after multiple injections (via enSERT) were not included in the downstream analyses.

[Reviewer 2 continued, additional comment] 1. Figure 1d is missing the proportion label for the 3+ marks category (shown in gray).

We have updated Figure 1d to include the proportion (50%) for the 3+ marks category.

[Reviewer 2 continued, additional comment] 2. Figure 3a is also missing the proportion label for 3+ marks category (shown in gray).

We have updated Figure 3a to include the proportion (32%) for the 3+ marks category.

[Reviewer 2 continued, additional comment] 3. It is difficult to distinguish the titles for each set of alternate data from the subtitles in Figure 4c. Consider making the titles a larger font or highlighting the schematic (a-c) along with the relevant title.

We have updated the labeling and spacing in Figure 4c to improve readability.

[Reviewer 2 continued, additional comment] 4. The color scales in the Figure S5 and S6 heatmaps are difficult to parse when expression is high (especially Fig S6c). They would benefit from numeric annotations.

We have updated the y-axis (TPM) to range from 0 to 50. We have added numerical values to the heatmaps in Figures S5 and S7. Please note that Figure S6 is now Figure S7.

Reviewer 3

Reviewer #3 (Remarks to the Author):

In this study, the authors carry out a tiling transgenic approach in two mouse loci to systematic assay for enhancer activity, and then relate findings to ENCODE chromatin signatures in concordant tissues. They find that “14% of enhancers did not show canonical enhancer-associated chromatin signatures in the tissue in which they are active”. This is an interesting study that demonstrates the limitations of current epigenomic annotations.

We thank the reviewer for acknowledging the general conclusion of our work, *i.e.*, the limitations of current epigenomic enhancer discovery methods for comprehensively detecting all bona fide enhancer elements in the genome.

Perhaps the most important clarification to be made about this study is that it does not show that some enhancers do not actually have these chemical modifications, instead they have been unable to capture them in existing datasets – largely ENCODE and published experiments with E11.5 forebrain and cerebellum scATAC. The fact that they cannot be detected is important, and the most plausible explanations can be inferred from the studies they perform. For example, the modifications could be present in a small proportion of cells in dissected tissue, so that the signal to noise is insufficient to enable detection. This is consistent with several examples where the enhancer Lacz activity is clearly restricted to very few cells (e.g. Figure S6). Another plausible explanation is that ChIP-seq/ATAC data is not of sufficient quality, typically because of well-known challenges to obtain high quality data from dissected mouse embryonic tissue. In fact, the Chip-seq data shown in figures 2 and 3 shows considerable background (which is quite usual in data that is obtained from ultra small tissue). In the case of scATAC, it is unclear if the coverage is truly exhaustive for less prevalent cell types, or how many accessible sites might not show signal above background yet still lacking statistical significance. The authors carefully examined earlier (E10.5) timepoints to rule out residual Lacz activity from a preceding stage, but the same caveats apply here: those assays in E10.5 tissues can miss activity from few cells, or because of the same type of technical limitations outlined above.

Finally, the authors find that the vast majority of the enhancers that “lack” active enhancer chromatin signatures actually do have enhancer activity marks in another tissue (although that is also seen for inactive DNA elements, so in the end there is only some enrichment, unclear if significant, in active elements).

Every one of the analyses they came up with effectively reduced the burden of hidden examples, which suggests that the residual number of hidden enhancers would be further reduced if they had access to more types of datasets: much larger and robust single cell datasets, ChIP-seqs with more pure tissues, perhaps even robust DNA methylation data, etc. A significant message of this study, therefore, is that it highlights the limitations of

mouse ENCODE data and calls for the generation of more extensive high quality datasets.

In conclusion, the authors state that they have identified “enhancers that are hidden from proper tissue-specific genome annotation using current chromatin-based enhancer identification approaches”, and “a substantial proportion of in vivo enhancers are hidden from discovery by conventional chromatin profiling methods”. But it this seems to be more accurately “using current chromatin-based enhancer datasets” rather than the approaches or methods themselves. It is very likely that more exhaustive, higher quality datasets using the same methods do reveal active enhancer signatures.

This distinction is interesting because previous studies have suggested that some enhancers have non-canonical modified histone marks. Furthermore, whether some enhancers might intrinsically not exhibit accessible chromatin in current assays (ATAC, DNase1). All of this is intriguing but is untested by the current study. The abstract, therefore, needs to be adapted to fully clarify that the reason why some enhancers are not detected is a limitation of current datasets, rather than the methods per se.

We agree with the reviewer. While we cannot exclude the possibility that subsets of these hidden enhancers are associated with hitherto undiscovered, non-canonical chromatin marks, it is very likely that the majority of hidden enhancers can be explained by insufficient signal-to-noise in small subsets of cells, by general sensitivity/quality issues of currently available datasets, or by lack of data for relevant tissues and timepoints. In the Abstract we have now better highlighted that the most likely explanation for the majority of hidden enhancers is of a technical, rather than a biological nature. In addition, in agreement with this reviewer’s comments, we emphasize that the limited quality of currently available datasets rather than the methodologies applied are the most likely reason for the absence of canonical enhancer marks at hidden enhancers.

We have updated the Abstract as follows:

Here we show that a substantial proportion of in vivo enhancers is hidden from discovery by the use of presently available chromatin-based datasets available for a wide range of embryonic tissues.

Our findings both suggest the existence of tens of thousands of enhancers throughout the genome that are absent in currently available chromatin-based datasets and underscore the growing utility of incorporating complementary and multimodal data for enhancer detection.

We have updated the Discussion as follows:

However, we also show that reliance on currently available chromatin-based datasets to identify candidate enhancers misses a notable portion of seemingly hidden enhancers that are active in the transgenic in vivo reporter assay but do not show any of the noted marks.

With some estimates of hundreds of thousands to on the order of one million candidate enhancers in mammalian genomes, one might speculate from our tiling study that there are tens of thousands of additional enhancers unaccounted for by current genome-wide chromatin catalogs.

Please also see our response to Reviewer 1, Point 5.

Another major point is that in the end the fraction of “active enhancers” that lacks marks is not so high. An alternative explanation is that they are false positives of this artificial enhancer assay, rather than false negatives of the chromatin marks. Unless the authors feel that this conundrum can be resolved, they may wish to incorporate this possibility to the conclusion.

We thank the reviewer for this suggestion and have added the following to the Discussion:

Although we considered additional chromatin data to identify hidden enhancers, we cannot exclude the possibility that a portion of these active enhancers could instead represent false positives in the transgenic reporter assay rather than false negatives originating from chromatin-based enhancer-prediction datasets.

Other specific points

- As outlined above, some examples might be false positives rate of this artificial assay, which does not take into account how enhancers might actually work in their native genomic sites. While this reviewer cannot think of a simple gold standard method to benchmark the method, much more information on replication of enhancer assays concerning expression patterns is essential to assess these findings. This should be clear in the main text and methods. So far the information that is provided is largely qualitative, how consistent are the expression patterns. Can authors provide more compelling data than just showing a handful of examples?

We thank the reviewer for their suggestion and have added the following to the Methods:

Tested tiling elements with reproducible tissue-specific enhancer-reporter activity (as determined by a panel of researchers) in at least two independent embryos and with at least one site-specific integration were reported as positive for the given tissue.

We note here that our actively maintained VISTA Enhancer Browser (<https://enhancer.lbl.gov>) contains both mouse *in vivo* results (including replicates for each element with tissue-specific enhancer activity identified) and their reproducibility from both the VISTA retrospective and tiling studies. We have updated Figure S7 (formerly Figure S6) to include the reproducibility of each depicted transgenic result, *i.e.*, the number of independent embryos with *LacZ*

staining in the considered tissue over the total number of transgenic embryos obtained.

- The overlaps with other tissues is interesting, but given the “background overlap” it would be more informative to ask if there are overlaps with snATAC specifically from other neural lineage clusters, where the priors for overlaps being meaningful are higher, and with more strict spatial resolution criteria.

In response to Reviewer 2, Point 1, we have provided additional information regarding hidden enhancers and tissue overlaps (see above). If this reviewer had additional analyses in mind, we would appreciate further clarification regarding the specific preexisting data sets and types of intersections suggested.

- The authors have studied mainly mouse ENCODE, and two scATAC datasets. Have they considered using other datasets? If no other valid dataset exists this should be explained.

The mouse developmental chromatin catalog from the latest ENCODE consortium data release¹⁸ was generated from mouse tissues collected and dissected by our group. Thus we are deeply familiar with how particular tissues were collected for the chromatin-based assays, and the potential correspondence of assayed tissues with the observed tissue-specific enhancer-reporter activity in our mouse *in vivo* assay¹⁹. Additionally, the chromatin data were analyzed using rigorously tested and publicly available ENCODE processing pipelines that enable chromatin state comparisons across both developmental stages and different tissues¹⁸. While we acknowledge that there are multiple other chromatin datasets available in similar mouse tissues, it is beyond the scope of this study to include these in the relevant analyses.

- Discussion: “we could not identify sequence, genomic, or other epigenomic properties that could distinguish hidden enhancers from their marked counterparts”. Please explain this sentence, it seems to me that the authors do claim that there are epigenomic differences.

We have updated the above sentence in the Discussion to the following:

However, apart from the absence of enhancer-associated chromatin marks, we could not identify sequence, genomic, or other epigenomic properties that could distinguish hidden enhancers from their marked counterparts.

Reviewer 4

Reviewer #4 (Remarks to the Author):

In this manuscript, Mannion, Visel, Pennacchio, and collaborators combine *in vivo* functional genomic approaches with ChIP-seq and ATAC-seq data to interrogate the

correlation between identification of enhancers using classic chromatin features (i.e. H3K4me1, H3K27ac, and ATAC-seq) versus functional identification using in vivo transgenic enhancer reporter assays.

The authors conclude that using H3K4me1, H3K27ac, and ATAC-seq is insufficient for a comprehensive annotation of enhancers in the mouse genome. To prove this, they compare enhancer identification by combining stage and tissue-matched H3K4me1 and H3K27ac ChIP-seqs and ATAC-seq with identification based on in vivo reporter assays. Some of these in vivo reporter assays were already present in their VISTA Enhancer Database and others have been performed to address the hypotheses in this manuscript.

A preliminary “retrospective” search for classic chromatin features on functionally validated enhancers from the VISTA database indicated that hidden enhancers comprise from 9 to 25% of all the VISTA database enhancers. This observation motivated the authors to analyze further if the existence of hidden enhancers is a widespread feature of the mouse genome. To assess this, they divided the Gli3 and Smad3/Smad6 genomic landscapes into 5 kb bins, clone these DNA regions into reporter plasmids and performed transgenic in vivo reporters assays. The authors observed that 26% of the ~5kb tiles that were positive in the in vivo functional assays lacked enhancer classical marks in stage and tissue-matched NGS data.

Then, in figure 3, the authors utilize non-classical enhancer histone marks to evaluate if hidden enhancers display any of these features. Their data shows that hidden enhancers do not contain any of these features (H3K27me3, H3K36me3, H3K4me2, H3K4me3, H3K9ac/H3K9me3). Next they compare other enhancer features such as evolutionary conservation, transcription factor binding sites, and presence of transposable elements and they conclude that none of these features distinguishes “canonical” from “hidden” enhancers.

Finally, the authors consider alternative hypotheses that could explain the lack of classic enhancer chromatin features in hidden enhancers. Their results conclude that some of the hidden enhancers at E11.5 displayed classical enhancer features at E10.5, thus the positive result in the transgenic reporter assays is likely due to residual lacZ reporter activity from activation in an earlier developmental stage. They also draw upon available snATAC-seq to test if tissue-heterogeneity contributes to the lack of chromatin accessibility in the bulk experiments and use human epigenomic data of syntenic regions to try to predict those hidden enhancers. Interestingly, the authors report that ENCODE cis-regulatory elements catalog that represents multiple cell lines and tissues does identify ~90% of the hidden enhancers.

Overall, the results presented in this manuscript confirm observations made by many in the field. Using classical marks is a good approximation for enhancer identification, but their comprehensive annotation in a cell and tissue-specific manner requires functional assessment in the precise chromatin environment (i.e. CRISPR-Cas9 deletion, dCas9-KRAB repression, reporter assays). From my point of view, this manuscript lacks the

scientific novelty expected from a Nature Communications research article. The existence of hidden enhancers is not sufficiently supported by the data. For this reviewer, the data only supports the lack of accurate measurements of chromatin features in cell types that may be rare in nature and reflects the exquisite spatial and temporal control of gene expression by enhancers. Alternatively, most of their lacZ reporter positive signals may be the product of an earlier developmental activation of the reporter, and this could explain the lack of correlation with the chromatin features. Last, selection of H3K4me1, H3K27ac, and accessibility as main chromatin features seems rather outdated. Nonetheless, this reviewer would like to remark that the functional assessment of the tiled Gli3 and Smad3/Smad6 genomic landscapes by in vivo reporter assays is an elegant approach and as they have included the results in the VISTA database, it will be a useful resource for scientists in the field.

We thank the reviewer for appreciating the scale of this study and the utility of the resulting resource.

We agree that it is well known among groups that use epigenomic methods to predict enhancers in the genome that these methods have imperfect specificity and sensitivity. However, the vast majority of such reports are anecdotal, and many negative results go unpublished, therefore it is difficult to assess the true extent of such limitations from currently available data. We believe that our systematic assessment of extended genomic regions to measure these effects more quantitatively provides major value for understanding the extent of the phenomenon of hidden enhancers, even if the most likely explanation in most cases is related to effects such as sensitivity, minor timepoint deviations, or restriction of activity to rare cell types.

In response to comment 1 of Reviewer 1, we have now also integrated a CRISPR/Cas9 deletion approach in mouse embryos to analyze the functional contribution of hidden enhancer elements in the *Gli3* regulatory landscape (new Fig. S8), adding further scientific novelty and complementing our systematic *in vivo* evaluation of the transcriptional enhancer landscapes of the *Gli3* and *Smad3/6* loci.

Please see below for additional comments on the choice of chromatin marks.

Major comments:

1-One of the potential explanations for the lack of classic histone modifications and accessibility at hidden enhancers could be tissue heterogeneity. H3K27ac, H3K4me1, and ATAC-seq assays were performed in bulk tissue, and thus they reflect enrichment across the population of cells. If a cell-type is scarce, it could still test positive in transgenic assay while the overall background would hinder the enrichment by ChIP-seq or ATAC-seq. In my opinion, it is not sufficiently proven that these enhancers do not exhibit chromatin marks, thus limiting the interpretation of the main conclusions of this manuscript (i.e. classic histone marks and accessibility are not enough for identifying enhancers).

We agree that some hidden enhancers may not be detected from bulk tissue-derived data due to activity restrictions to small subsets of cells. This notion is directly supported by our finding that the inclusion of single-cell data (where available) correctly identifies additional enhancers that were missed by bulk tissue-derived data. While primarily technical in nature, we believe that our chromatin/epigenomic analysis data provides an important and direct demonstration of this effect.

We have now described this more clearly in the Discussion as follows:

While public single cell chromatin accessibility data are currently available for a few tissues in mice^{20,21}, it is likely that additional developing tissues (e.g., face, limb, heart) will soon be surveyed at single-cell resolution. As supported by our comparisons of hidden forebrain and hindbrain enhancers, these single-cell approaches should enable the resolution of both common and rare cell types in tissues and, subsequently, the identification of enhancers missed by bulk tissue-derived data.

Please also see our response to Reviewer 1, Point 5.

2-The authors partially address major comment 1 by analyzing snATAC-seq of forebrain and hindbrain in Figure 4B, and they report enrichment of ATAC-seq signal in 8 enhancers, 2% of the total hidden enhancers. This percentage is misleading and it should refer to the total of hidden forebrain and hindbrain enhancers (n=44). With this calculation 18% of forebrain and hindbrain hidden enhancers display classic enhancer ATAC-seq signal when evaluated in a single cell setting.

We thank the reviewer for this important observation. We have updated the relevant sentence in Results as follows:

Of the 44 hidden enhancers that are active in either the forebrain or the hindbrain, only 8 (18%) could be identified via available corresponding single cell data (**Fig. 4b**).

Minor comments: 1-The Vista Database candidate enhancers were selected by means of p300 enrichment or evolutionary conservation. It would be useful to know how many of the hidden enhancers were in fact originally identified by using p300 ChIP-seq.

The VISTA Enhancer Browser contains a collection of enhancers originally predicted by a wide range of methods, including various types of comparative genomics (different tools and phylogenetic distances), epigenomic marks (p300, known enhancer-associated histone marks, candidate enhancer-associated proteins and histone marks), experimental data from MPRA, hypothetical enhancers identified in gene-centric studies, and many more. In many cases, multiple signals (e.g., chromatin marks and conservation) were combined to predict enhancers. Thus, any retrospective analysis with the VISTA database alone can potentially be misleading because the tested sequences are highly

biased towards positive enhancers (relative to randomly selected non-coding sequences). These considerations were the driver for our effort to perform tiling in an unbiased manner across an extended genomic interval, *i.e.*, the core data set of the present study.

A minority of the enhancers in the VISTA Enhancer Browsers were first identified using p300. Of the 1,272 positive VISTA enhancers analyzed in the present study, only 300 (23%) were first tested based on p300 signal as the primary mark. Of the 309 tissue-specific hidden enhancers in this study, only 5 could be identified using previously generated p300 ChIP-seq peaks from human and mouse tissues^{10,22-24}.

2-In my opinion, the nomenclature “hidden” enhancers is misleading, as it invokes the idea of a novel class of enhancers. The authors use this word to distinguish hidden enhancers from enhancers marked by classical enhancer features, but for this reviewer, the lack of chromatin marks in the hidden enhancers has not been sufficiently proven in this manuscript. Additionally, one could argue that their vision of classical marks is outdated. Researchers these days combine H3K4me1/H3K27ac/ATAC-seq, transcription factor binding sites clusters, RNAPII presence, 3D-chromatin measurements etc. to identify enhancers.

We thank the reviewer for their critical assessment of our terminology. However, we believe that “hidden” is an appropriate term to describe the observed effects. Merriam-Webster's dictionary defines “hidden” as “being out of sight or not readily apparent”, which is in our opinion an accurate and appropriate description of the properties of these sequences. Just like for any hidden object, this does not imply that they have any properties fundamentally distinct from non-hidden objects, nor that they cannot be discovered using different methods. We directly demonstrate how the consideration of additional data types can reveal some but not all of these previously hidden enhancers.

We agree that many other marks have been found to be enriched at enhancers. However, based on our extensive experience in evaluating such marks for the prediction of *in vivo* enhancers, H3K27ac in combination with accessible chromatin (*e.g.*, from ATAC-seq) continues to provide the most robust combination of specificity and sensitivity at this point. We demonstrated this in extensive benchmarking studies of epigenomic data vs. *in vivo* data as part of the ENCODE project^{18,25}.

As described above in response to comments by Reviewer 3, we have now also clarified in the manuscript that it is likely that the majority of the observed hidden enhancers are explained by insufficient signal-to-noise in small subsets of cells, by general sensitivity/quality issues of currently available datasets, or by lack of data for relevant tissues and timepoints.

3-Figure 1c. is missing the scale of the signal.

We have updated Figure 1c to include the signal (y-axis) for each assay.

4-Authors should indicate what is the meaning of the numbers in the chromatin features columns of Supplementary Table 2.

We have added the following to the “Chromatin intersections and hidden enhancer identification” Methods section:

We record in **Table S2** the total number of peaks (for a given chromatin dataset) that overlap with each tested element.

5-Specific command used for intersecting chromatin features and enhancers using bedtools should be detailed, as too stringent colocalization requirements may underestimate the number of canonical enhancers.

Please see our response to Reviewer 2, Point 5.

6-Authors conclude that 56% of the initial hidden enhancers could in fact be identified using chromatin features from E10.5, suggesting that the lacZ activity is residual from an earlier activation. This reviewer positively appreciates this analysis and suggests that more details about the pitfalls of transgenic *in vivo* assays could be provided in the discussion.

We have added the following sentences to the discussion:

Enhancer-associated chromatin marks at an earlier stage (E10.5) identified the largest proportion (56%) of E11.5 hidden enhancers. Reporter systems that provide a higher resolution of *in vivo* temporal readout of enhancer-reporter activity than the utilized LacZ-based transgenic assay, or functional deletion/inhibition experiments, will determine whether this portion of hidden enhancers exhibit largely stage-restricted functions. Lack of endogenous chromatin context represents another limitation of using site-directed transgenic reporter assays for evaluation of enhancer activity and function.

7-To rule out the possibility of hidden enhancers being marked by “non-classical” histone marks, the authors use ENCODE data of K27me3, K36me3, K4me2/3, K9ac, and K9me3. Results in Figure 3b and Supplementary Figure 7 conclude none of these marks are present at the hidden enhancers. This is not explained in the text at all.

We have added the following sentence to the main text:

Aside from a lack of the three enhancer-associated chromatin marks, we found a majority of these hidden enhancers also were without alternative histone marks examined by ENCODE (H3K27me3, H3K36me3, H3K4me2, H3K4me3, H3K9ac, H3K9me3) (**Figs. 3b, S6**).

Fig. S7 was relabeled to Fig. S6 (and vice-versa) to accommodate this new sentence.

8-In the search for differences between canonical and hidden enhancers, the authors compare enriched TFBSs concluding that there are not particular motifs enriched. As it is, this analysis is not meaningful unless the hypothesis is that hidden enhancers are a

novel class of enhancers regulated by a common set of TFs across tissues. In my opinion, this analysis would bring more utility when performed in a tissue-by-tissue fashion, where presence or absence of a motif in a hidden enhancer could provide relevant biological information (for example reveal a temporal aspect of the activation of the enhancer that could hint reasons why there is a disconnection between chromatin features and in vivo activity).

We apologize for the confusion created by our description of this study. Our assumption (and stated conclusion) is that hidden enhancers, beyond our ability to detect them using available methods and datasets, do NOT represent a functionally distinct, novel class of enhancers. We performed this analysis to test the alternative hypothesis, *i.e.*, the possible presence of differences in TFBS content between hidden and non-hidden enhancers. The absence of enrichment of any particular TFBS disproves this alternative hypothesis, providing additional support (though not definitive proof) for our conclusion that no functional differences exist.

We used HOMER²⁶ to evaluate on a per-tissue basis whether hidden enhancers had any enrichment of TF motifs relative to marked enhancers (*e.g.*, whether or not hidden forebrain enhancers had specific TF motifs that distinguished them from marked forebrain enhancers). From these tissue-by-tissue comparisons, potentially enriched TF motifs were reported only in hidden forebrain and hidden limb enhancers (see Table S5. Summary of hidden enhancer transcription factor motif analysis). No enriched TF motifs were reported for the remaining four tissues.

Furthermore, the Methods section has been updated as follows:

HOMER²⁶ (v4.10) was used to assess, per tissue, the enrichment of both known and de novo motifs in hidden enhancers relative to their marked counterparts, via *findMotifsGenome.pl* and the following parameters: `-size given -len 8,9,10,12,14 -bg <background file = hidden + marked enhancers>`.

References

1. Osterwalder, M. *et al.* Enhancer redundancy provides phenotypic robustness in mammalian development. *Nature* **554**, 239–243 (2018).
2. Dickel, D. E. *et al.* Ultraconserved Enhancers Are Required for Normal Development. *Cell* **172**, 491-499.e15 (2018).
3. Osterwalder, M. *et al.* HAND2 targets define a network of transcriptional regulators that compartmentalize the early limb bud mesenchyme. *Dev. Cell* **31**, 345–357 (2014).
4. Lopez-Rios, J. *et al.* GLI3 constrains digit number by controlling both progenitor proliferation and BMP-dependent exit to chondrogenesis. *Dev. Cell* **22**, 837–848 (2012).
5. Levin, M. Left-right asymmetry in embryonic development: a comprehensive review. *Mech. Dev.* **122**, 3–25 (2005).
6. Long, H. K. *et al.* Loss of Extreme Long-Range Enhancers in Human Neural Crest Drives a Craniofacial Disorder. *Cell Stem Cell* **27**, 765-783.e14 (2020).
7. Pradeepa, M. M. *et al.* Histone H3 globular domain acetylation identifies a new class of enhancers. *Nat. Genet.* **48**, 681–686 (2016).
8. Narita, T. *et al.* Acetylation of histone H2B marks active enhancers and predicts CBP/p300 target genes. *Nat. Genet.* **55**, 679–692 (2023).
9. Rada-Iglesias, A. *et al.* A unique chromatin signature uncovers early developmental enhancers in humans. *Nature* **470**, 279–283 (2011).
10. Visel, A. *et al.* ChIP-seq accurately predicts tissue-specific activity of enhancers. *Nature* **457**, 854–858 (2009).
11. Calo, E. & Wysocka, J. Modification of enhancer chromatin: what, how, and why? *Mol. Cell* **49**, 825–837 (2013).
12. Nord, A. S. & West, A. E. Neurobiological functions of transcriptional enhancers. *Nat. Neurosci.* **23**, 5–14 (2020).
13. Abassah-Oppong, S. *et al.* A gene desert required for regulatory control of pleiotropic *Shox2* expression and embryonic survival. *Nat. Commun.* **15**, 8793 (2024).
14. Dixon, J. R. *et al.* Topological domains in mammalian genomes identified by analysis of chromatin interactions. *Nature* **485**, 376–380 (2012).
15. Bonev, B. *et al.* Multiscale 3D Genome Rewiring during Mouse Neural Development. *Cell* **171**, 557-572.e24 (2017).
16. Chen, Z. *et al.* Increased enhancer-promoter interactions during developmental enhancer activation in mammals. *Nat. Genet.* **56**, 675–685 (2024).
17. Kvon, E. Z. *et al.* Comprehensive In Vivo Interrogation Reveals Phenotypic Impact of Human Enhancer Variants. *Cell* **180**, 1262-1271.e15 (2020).

18. Gorkin, D. U. *et al.* An atlas of dynamic chromatin landscapes in mouse fetal development. *Nature* **583**, 744–751 (2020).
19. Visel, A., Minovitsky, S., Dubchak, I. & Pennacchio, L. A. VISTA Enhancer Browser—a database of tissue-specific human enhancers. *Nucleic Acids Res.* **35**, D88-92 (2007).
20. Preissl, S. *et al.* Single-nucleus analysis of accessible chromatin in developing mouse forebrain reveals cell-type-specific transcriptional regulation. *Nat. Neurosci.* **21**, 432–439 (2018).
21. Sarropoulos, I. *et al.* Developmental and evolutionary dynamics of cis-regulatory elements in mouse cerebellar cells. *Science* **373**, (2021).
22. Blow, M. J. *et al.* ChIP-Seq identification of weakly conserved heart enhancers. *Nat. Genet.* **42**, 806–810 (2010).
23. May, D. *et al.* Large-scale discovery of enhancers from human heart tissue. *Nat. Genet.* **44**, 89–93 (2011).
24. Attanasio, C. *et al.* Fine tuning of craniofacial morphology by distant-acting enhancers. *Science* **342**, 1241006 (2013).
25. ENCODE Project Consortium *et al.* Expanded encyclopaedias of DNA elements in the human and mouse genomes. *Nature* **583**, 699–710 (2020).
26. Heinz, S. *et al.* Simple combinations of lineage-determining transcription factors prime cis-regulatory elements required for macrophage and B cell identities. *Mol. Cell* **38**, 576–589 (2010).

Summary Response to Reviewers Comments

NCOMMS-23-00294-T: Uncovering Hidden Enhancers Through Unbiased In Vivo Testing

We thank the reviewers for their comments. Below we have summarized our responses to the points raised and changes to the manuscript.

Reviewer #1 (Remarks to the Author):

The authors have responded thoroughly to the reviewers questions, suggestions, and critiques.

This is an important body of work that will benefit those searching for tissue-specific enhancers. I am pleased to see that they were able to correlate function with transgenic activity of two hidden enhancers. This is a significant amount of work, and I believe their diligence has made this an even more important and citable manuscript. I realize that redundancy can obscure the functional contributions of enhancers and I applaud the authors for navigating through this challenge using a compromised genetic (*Gli3^d*) background to accentuate the contribution of these hidden enhancers in anterior limb patterning. Excellent work!!

I do have one critique/recommendation. The authors describe a CRISPR/cas9 deletion upstream of *Gli3*. I think this deletion is named iTAD by the authors but it is not defined in the results section. When the abbreviation shows up later in the paragraph – it is again not defined. I would ask the authors to define iTAD for the readers, probably naming the deletion when it is first described in the results. Later in the discussion this deletion is again described but not named. “deletion of a nearly 300kb *Gli3*-upstream genomic interval containing multiple canonical and designated hidden enhancer elements” It would benefit to the reader to have the name again attributed to this deletion by adding in parentheses iTAD (iTAD).

Looking forward to referencing this paper in the future!
All the best, Kerby Oberg

We thank the reviewer for their positive feedback and their appreciation of the corresponding work. We appreciate the recommendation to clarify terminology and have updated the relevant texts as follows:

Results: Additionally, using CRISPR/Cas9 genome editing, we generated a mouse line harboring a deletion of a 303kb intra-TAD (iTAD) region upstream of the *Gli3* TSS. The deleted region (iTAD^d) encompasses seven canonical and four hidden enhancers, each with distinct tissue-specific activities (Fig. S8A).

Discussion: Remarkably, the deletion of a 303kb intra-TAD (iTAD) *Gli3*-upstream region containing multiple canonical and designated hidden enhancer elements with tissue-specific activities did not result in overt phenotypic alterations, indicating the presence of multiple tissue-specific redundant enhancers outside of the deleted region.

Reviewer #2 (Remarks to the Author):

I thank the authors for their clear and complete responses to the reviews. I believe the manuscript is much clearer now and can be accepted. I only have 2 minor comments:

- There is a typo in H3K27ac in line 381 (Discussion).
- In Line 394 the phrase: "...we can expect to better understand and the unique characteristics of hidden enhancers...". However, in the paper you do not find strong evidence that these sequences are functionally different from marked enhancers (outside of their identification strategy). This sentence could be clarified if you do not think that hidden enhancers are a separate class of element (which seems to be the case given the rest of the paper and response to reviewers).

We thank the reviewer for their positive feedback and for identifying these corrections in the manuscript. We have updated the following texts in the Discussion section:

We were unable to find any distinguishing properties of hidden enhancers using additional histone modifications profiled in the ENCODE developmental chromatin catalog. In this study we could not systematically exclude in a tissue- and developmental stage-matched manner the possibility that other histone modifications^{50,51} (e.g., H3K27ac; H2BNTac) or proteins^{22,52,53} (e.g., p300/CBP, Pol II) are present at these enhancers.

As sequencing expands to cover the full range of human tissues, diversity, environmental perturbations, and as related technologies provide even higher resolution approaches to probe gene regulatory activity, we can expect to better understand and annotate the characteristics of enhancers and their functional significance in transcriptional regulation.

Reviewer #4 (Remarks to the Author):

The authors have addressed all my concerns. I appreciate the nuance regarding the technical limitations of detecting the "hidden" enhancers and the clarifications regarding their "technical" versus "biological" nature.

I support the publication of the article as it is.

Congratulations to the team and thank you for your valuable resources for the enhancer field!

We thank the reviewer for their positive feedback and their appreciation of the contributions of this work.